# Maladaptive positive feedback production of ChREBPβ underlies glucotoxic β-cell failure

Liora S. Katz [1], Gabriel Brill[2], Pili Zhang[1], Anil Kumar[3], Sharon Baumel-Alterzon[1], Lee B. Honig[1], Nicolás Gómez-Banoy[4], Esra Karakose[1], Marius Tanase[1], Ludivine Doridot[5], Alexandra Alvarsson[1,6], Bennett Davenport[7], Peng Wang [1], Luca Lambertini [1], Sarah A. Stanley [1], Dirk Homann [1], Andrew F. Stewart [1], James C. Lo [4], Mark A. Herman [8,9], Adolfo Garcia-Ocaña[1] & Donald K. Scott [1] ✉

Preservation and expansion of β-cell mass is a therapeutic goal for diabetes. Here we show that the hyperactive isoform of carbohydrate response-element binding protein (ChREBPβ) is a nuclear effector of hyperglycemic stress occurring in β-cells in response to prolonged glucose exposure, high-fat diet, and diabetes. We show that transient positive feedback induction of ChREBPβ is necessary for adaptive β-cell expansion in response to metabolic challenges. Conversely, chronic excessive β-cell-specific overexpression of ChREBPβ results in loss of β-cell identity, apoptosis, loss of β-cell mass, and diabetes. Furthermore, β-cell "glucolipotoxicity" can be prevented by deletion of ChREBPβ. Moreover, ChREBPβ-mediated cell death is mitigated by over-expression of the alternate CHREBP gene product, ChREBPα, or by activation of the antioxidant Nrf2 pathway in rodent and human β-cells. We conclude that ChREBPβ, whether adaptive or maladaptive, is an important determinant of β-cell fate and a potential target for the preservation of β-cell mass in diabetes.

All major forms of diabetes arise from insufficient β-cell mass. Thus, extensive research efforts are underway to either preserve or expand functional β-cell mass[1,2]. Under physiological conditions, increased blood glucose acts as a systemic adaptive β-cell mitogen, expanding functional β-cell mass through the proliferation of β-cells to meet demand for insulin[3]. However, prolonged hyperglycemia gives rise to glucose toxicity, which impairs insulin production and secretion, promoting a vicious cycle with ever-increasing glucose concentrations and ever-declining β-cell function, and eventually β-cell death[4]. Thus, depending on concentration and duration, increased blood glucose may drive either adaptation or diabetic pathogenesis.

Carbohydrate response-element binding protein (ChREBP, gene name *MLXIPL*) is a glucose-responsive transcription factor, originally cloned from liver, but also expressed at similar levels in pancreatic β-cells[5,6]. A major target gene of ChREBP is thioredoxin interacting protein (Txnip), which binds to and inhibits the antioxidant enzyme, thioredoxin (Txn), thereby promoting oxidative damage in β-cells. Depletion of ChREBP prevents, and overexpression of Txnip promotes glucose toxicity and glucose-mediated β-cell death[7]. In contrast to these observations, our group found that ChREBP is necessary for glucose-stimulated β-cell proliferation, a critical process for adaptive expansion of β-cells. Furthermore, overexpression of the full-length

[1]Diabetes, Obesity and Metabolism Institute, Icahn School of Medicine at Mount Sinai, One Gustave L. Levy Place, Box 1152 New York 10029, USA. [2]Pharmacologic Sciences Department, Stony Brook University, Stony Brook, NY, USA. [3]Metabolic Phenotyping Core, University of Utah, 15N 2030 E, 585, Radiobiology building, Room 151, Salt Lake City, UT 84112, USA. [4]Weill Center for Metabolic Health and Division of Cardiology, Department of Medicine, Weill Cornell Medicine, New York, NY 10021, USA. [5]Institut Cochin, Université de Paris, INSERM, CNRS, F-75014 Paris, France. [6]Alpenglow Biosciences, Inc., 98103 Seattle, WA, USA. [7]12800 East 19th Ave, Anschutz Medical Campus, Room P18-9403, University of Colorado, Aurora, CO 80045, USA. [8]Division of Endocrinology and Metabolism and Duke Molecular Physiology Institute, Duke University Medical Center, Durham, NC, USA. [9]Section of Diabetes, Endocrinology, and Metabolism, Baylor College of Medicine, One Baylor Plaza, MS: 185, R614, 77030 Houston, TX, USA. ✉e-mail: donald.scott@mssm.edu

form of ChREBP, now known as ChREBPα, does not cause β-cell death, but rather augments glucose-stimulated β-cell proliferation[5,8]. These apparently disparate observations may be explained by expression and function of the different isoforms of ChREBP.

There are two major isoforms of ChREBP, ChREBPα and ChREBPβ. Both are expressed in β-cells, as they are in adipocytes and hepatocytes[9,10]. ChREBPβ mRNA is transcribed from an alternative promoter located ~17,000 bp upstream of the ChREBPα transcription start site and is alternatively spliced and translated so that ChREBPβ lacks the low glucose inhibitory domain (LID) and nuclear export signal (NES) of the canonical full-length ChREBPα (Supplementary Fig. 1a). Thus, ChREBPβ is constitutively active and nuclear, and orders of magnitude more transcriptionally potent than ChREBPα[11]. Importantly, ChREBPβ expression is driven by powerful carbohydrate response elements (ChoREs) located near its own alternative transcription start site[10–12]. Therefore, increased glucose metabolism activates ChREBPα and, if sustained, initiates a vigorous positive feedback loop, with the newly synthesized ChREBPβ binding to its own ChoRE, producing more and more ChREBPβ (Supplementary Fig. 1b). Gain- and loss-of-function experiments in vitro and ex vivo revealed that the induction of ChREBPβ is the molecular engine that drives glucose-stimulated β-cell proliferation[10]. One goal of the present study was to test whether this process is necessary for the adaptive expansion of β-cells following a high-fat diet.

ChREBPβ is expressed at high levels in diabetes, and overexpression of ChREBPβ or a close homologue of ChREBPβ that lacks the low glucose inhibitory domain and is thus constitutively active results in cell death of cultured β-cells[10,13,14]. Thus, the molecular engine that drives glucose-stimulated β-cell transcription may also mediate cell death when levels of ChREBPβ get too high. By contrast, overexpression of ChREBPα does not cause β-cell death because it stimulates the Nrf2 antioxidant pathway, delivering protection from oxidative damage[8,15]. Despite these recent advances, key questions and controversies remain on the roles of the two ChREBP isoforms. For example, how is it that loss of ChREBP [presumably both isoforms[16]] and overexpression of ChREBPα[8] are both protective in settings of β-cell oxidative stress? Similarly, Shalev and colleagues found that acute overexpression of ChREBPβ decreases ChREBPα expression, and hypothesized that ChREBPβ is expressed at high levels in diabetes to block the presumed destructive effects of ChREBPα in diabetes[13], a model clearly different from the one described above.

Here we clarify the roles of the ChREBP isoforms in β-cells using a variety of tools and in vitro and in vivo approaches that distinguish the two isoforms. We find that under conditions of high concentrations of glucose, high-fat diet, or diabetes, ChREBPβ becomes the predominant isoform found in the nucleus of β-cells. Furthermore, we found that a modest, physiological induction of ChREBPβ is necessary for adaptive proliferation and expansion of β-cells after a high-fat diet. However, robust β-cell-specific overexpression of ChREBPβ mimics glucose toxicity in vivo, and results in frank diabetes. In addition, we find that deletion of ChREBPβ prevents β-cell glucolipotoxicity. Finally, we present novel approaches to mitigate the effects of ChREBPβ-mediated toxicity in rodent and human β-cells, providing insights for possible future therapies to preserve β-cell mass and combat type 2 diabetes. These observations place CHREBP isoforms at a nexus of control of the β-cell response to metabolic challenge and provide a revised understanding of glucolipotoxicity.

## Results

### ChREBPβ is the predominant nuclear isoform after prolonged exposure to high glucose

To understand the mechanisms and consequences of the dynamic relationship between activation of ChREBPα and the positive feedback induction of ChREBPβ (Supplementary Fig. 1), it was critically important to devise and use tools that distinguish the two isoforms (Supplementary Fig. 2). Since ChREBPβ is a truncated version of ChREBPα, and lacks the N-terminal low glucose inhibitory (LID) domain[11], an antibody directed against an N-terminal epitope identifies only ChREBPα, and an antibody directed against the C-terminal region recognizes both ChREBPα and ChREBPβ (Supplementary Figs. 1–3). We found ChREBP in the nucleus of INS-1-derived 832/13 cells (hereafter INS-1 cells), as expected, using an antibody recognizing the C-terminus and thus both isoforms after 24 h in 20 mM glucose (Fig. 1a), consistent with prior reports[17,18]. Interestingly, when we used an N-terminal antibody, ChREBPα appeared mostly cytoplasmic, with no discernable difference between low and high glucose (Fig. 1b). Thus, the predominant nuclear ChREBP isoform was ChREBPβ, which seemed at odds with the canonical view that ChREBPα translocates to the nucleus in response to increased glucose metabolism[19]. To confirm that ChREBPβ was the nuclear isoform, we employed islets from floxed ChREBPβ mice[10]. Dispersed islet cells transduced with either an adenovirus expressing LacZ, as a control, or Cre recombinase to remove ChREBPβ, were incubated in 6 or 20 mM glucose for 48 h. As seen in Fig. 1c, d, the N-terminal antibody (ChREBPα) was mostly cytoplasmic, even in 20 mM glucose, while the C-terminal antibody showed bright nuclear staining after culture in 20 mM glucose, indicating the nuclear presence of ChREBPβ. Importantly, after Cre-mediated deletion of ChREBPβ, there was no nuclear ChREBP immunolabeling. To examine this further, we performed a time course experiment with adenovirally-expressed Flag-tagged ChREBPα, and an antibody against the Flag epitope in INS-1 cells after exposure to 2 or 20 mM glucose (Fig. 1e, f). Flag-tagged ChREBPα was mostly cytoplasmic when cultured in 2 mM glucose. After replacing the media with 20 mM glucose, Flag-tagged ChREBPα migrated rapidly into the nucleus within 5 min, in concert with previous observations[17,18]. However, after 3 h in 20 mM glucose ChREBPα appeared cytoplasmic, and remained cytoplasmic at 6 and 24 h (Fig. 1f, g). These studies illustrate that ChREBPα moves transiently into the nucleus in response to glucose stimulation, but rapidly returns (minutes) to the cytoplasm. In contrast, ChREBPβ, induced by high glucose via ChREBPα, becomes both elevated and nuclear.

To determine whether ChREBPβ becomes the major isoform bound to DNA, we performed a time course chromatin immunoprecipitation (ChIP) experiment using antibodies directed against either the N-terminus (recognizing ChREBPα only) or the C-terminus (recognizing both isoforms; Supplementary Fig. 3). We selected the mouse Pklr promoter as a target because it contains a well-studied ChoRE[5,12,20,21]. The recruitment over time of ChREBPα to both the Pklr ChoRE region and the ChREBPβ (Mlxipl) ChoRE region increased for the first 30 min, and sharply decreased and plateaued for the remainder of the experiment (Fig. 1h, i). The recruitment of both isoforms, as determined by the C-terminal antibody, increased and decreased, similar to the signal from the N-terminal antibody, but then increased for the duration of the 18 h experiment. These observations strongly suggested that recruitment of ChREBPβ continued throughout the time course of the experiment, whereas recruitment of the ChREBPα isoform increased only during the first 30 min of glucose treatment, broadly consistent with the experiments in Fig. 1a–f. These results are consistent with a model in which ChREBPβ becomes the major nuclear ChREBP isoform bound to DNA after prolonged exposure to high concentrations of glucose. Together, these observations strongly suggest that ChREBPα is transiently nuclear, and that ChREBPβ is the primary nuclear isoform after prolonged treatment with high concentrations of glucose.

An alternative explanation of the results in Fig. 1 could be that the epitope of the N-terminal antibody (recognizing only ChREBPα) becomes "masked" by the assembly of the transcriptional apparatus. In this scenario, the glucose-dependent recruitment of large co-activator protein complexes might sterically hinder the binding of antibodies, rendering ChREBPα "invisible". To address this possibility, we used CRISPR/Cas9 editing to add fluorescent tags to the 5′ and 3′ ends of

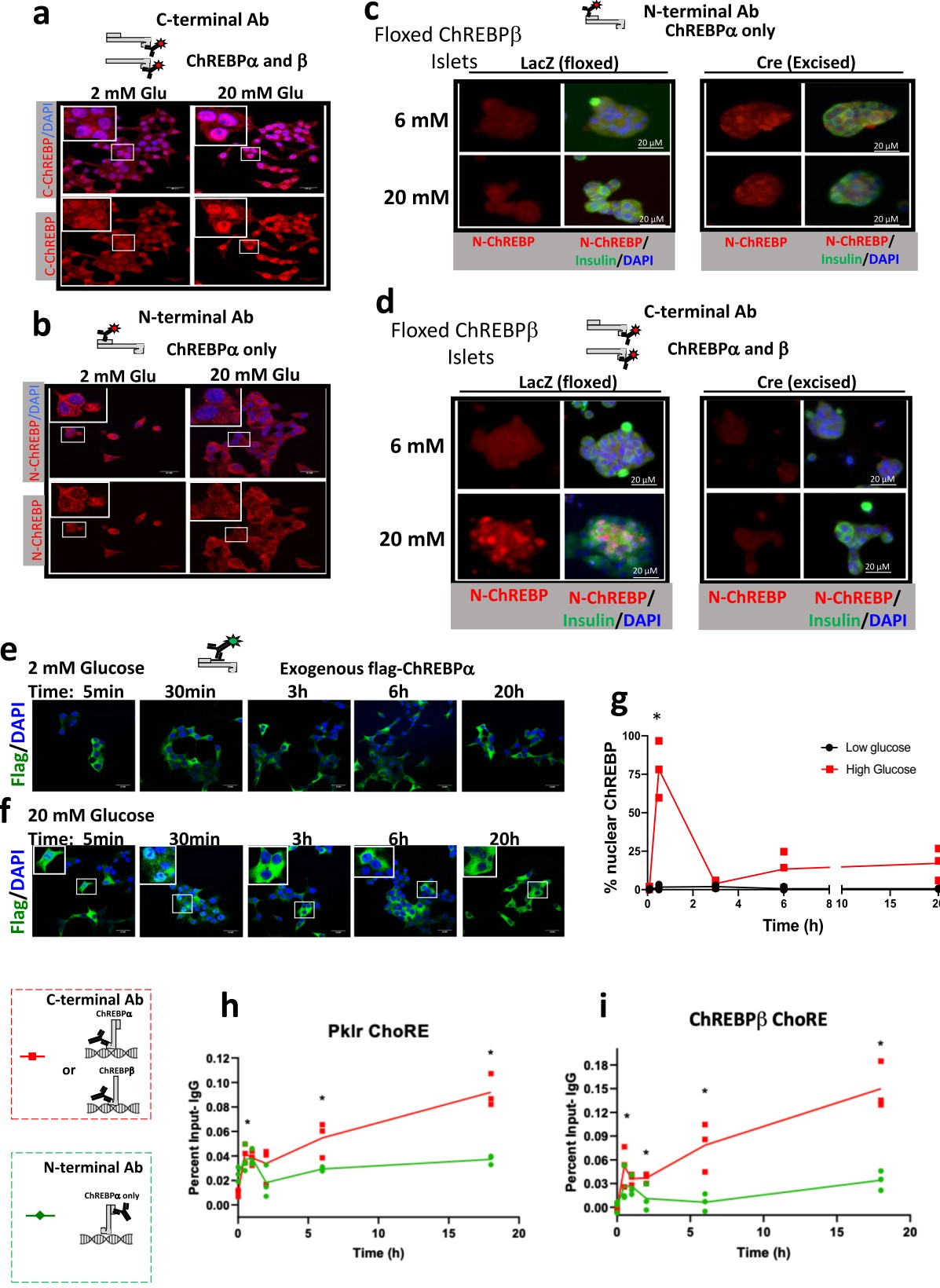

full-length ChREBPα in INS-1 cells. mCherry was attached to the N-terminal LID domain, which identifies ChREBPα exclusively (Red cells, see schematic Fig. 2a). We also generated cells labeled with eGFP on the C-terminus (Green cells), representing both ChREBPα and ChREBPβ, and doubly labeled cells (Red/Green cells), in which ChREBPα appeared red/green, and ChREBPβ appeared green. The cells were FACS sorted and validated via PCR, RT-PCR and Western blots (Supplementary Figs. 4–6). Red and Green cells were cultured in increasing concentrations of glucose (2–20 mM) for 72 h. Live-cell imaging demonstrated cytoplasmic localization for the ChREBPα regardless of glucose concentration, and nuclear localization for ChREBPβ at the highest concentration (Fig. 2b–d; Supplementary

**Fig. 1 | ChREBPβ is the main nuclear isoform after prolonged exposure to high concentrations of glucose. a, b** Ins-1 cells were cultured in low (2 mM) or high (20 mM) glucose for 24 h. Cells were fixed and immunostained with the C-terminus antibody for ChREBP, which recognizes both ChREBPα and ChREBPβ (**a**), or the N-terminus antibody for ChREBP, specific for ChREBPα (**b**). The results are representative of three independent experiments. **c, d** Islets from floxed ChREBPβ mice were isolated and transduced with adenoviruses expressing LacZ as a control, or Cre recombinase to excise ChREBPβ and cultured in 6 mM or 20 mM glucose. After 48 h, cells were fixed and immunostained with the N-terminal (**c**) or C-terminal (**d**) antibody; nuclei were stained with DAPI. Shown is a representative result of 3 independent experiments. **e, f** Ins-1 cells were transduced with an adenovirus

expressing flag-tagged ChREBPα and treated with 2 mM (**e**) or 20 mM glucose (**f**) for the indicated times. Cells were fixed and immunostained with an anti-Flag antibody and nuclei were stained with DAPI. The results from (**f**) were quantified in (**g**). The results are representative of three independent experiments. *, mean, $P < 0.05$. **h, i** Ins-1 cells were cultured overnight in 2 mM glucose and then glucose was added to a total of 20 mM for the indicated times; chromatin was isolated and processed for chromatin immunoprecipitation using antibodies recognizing the C-terminus, the N-terminus, or IgG. DNA was amplified by qPCR using primers specific for regions near the ChoREs on the *Pklr* (**h**), or *Mlxipl* (ChREBPβ) (**i**) gene promoters. The data are the means ± SEM of the percent input after subtraction of the IgG control. $n = 4$; *$p < 0.05$; ***$P < 0.0005$; ****$P < 0.0001$ using two-way ANOVA.

Fig. 7), independently confirming the results in the preceding paragraphs.

We next performed time-lapse confocal microscopy in INS-1 cells bearing fluorescent tags on both ends of ChREBP (Red/Green cells), while culturing the cells in 2 mM or 20 mM glucose, or after transitioning the cells from low to high glucose. We quantified the nuclear localization of red fluorescence (representing ChREBPα) and exclusively green fluorescence (representing ChREBPβ). After culture for 24 h in 2 mM glucose, nearly all of the fluorescence was cytoplasmic (Fig. 2e, f, left panels). Culture in 20 mM glucose for 24 h resulted in predominantly green nuclear fluorescence, indicating ChREBPβ as the principal nuclear form (Fig. 2e, f right panels). The green signal (ChREBPβ) was mostly nuclear throughout the time of acquisition. In contrast, the red signal (ChREBPα) was present in approximately 50% of the nuclei with a more random distribution, illustrating that ChREBP shuttles between the nucleus and the cytoplasm, and that increased glucose accelerates the rate of nuclear entry[17]. When the glucose in the medium was changed from 2 mM to 20 mM (Fig. 2e, f, middle panels), there was a rapid nuclear localization of ChREBPα (red and green fluorescence) followed by a separation of florescent signals after approximately 30 min, agreeing with our observations in Fig. 1 (and see Supplementary Fig. 8a and Supplementary Movie 1). Interestingly, we found that nuclear localization of either isoform required ongoing protein translation as it was inhibited by cycloheximide (Supplementary Fig. 8b). Together, these results suggested a model wherein ChREBPα is mostly cytoplasmic at low glucose, but with increased glucose metabolism, ChREBPα transiently becomes more nuclear to induce the production of ChREBPβ. ChREBPβ remains predominantly nuclear due to the absence of the LID domain (including the NES) and is more potently and constitutively active when compared to ChREBPα[11]. With sustained high levels of glucose, this positive feedback mechanism continues to produce more ChREBPβ, ensuring that ChREBPβ becomes the major isoform recruited to ChoREs (Supplementary Fig. 1b).

## ChREBPβ expression and nuclear localization correlate with hyperglycemia and diabetes

ChREBPβ was visualized in the nucleus of mouse β-cells in vivo, using our validated C-terminal antibody approach under quiescent, metabolically stressful, and diabetic conditions (Fig. 3a). The N-terminal antibody, representing ChREBPα only, was mostly cytoplasmic in all the conditions tested (Fig. 3a). In C57BL/6 mice, we observed a gradation of ChREBPβ expression: no detectable expression on a standard chow diet; modest abundance after one week on a high-fat diet, representing a normal physiological adaptive response[10], and very high expression levels in diabetic *db/db* mice (Fig. 3a). In addition, human islets labeled with a RIP-ZsGreen-expressing adenovirus were sorted to obtain pure β-cells[22]. The ratio of ChREBPβ to ChREBPα mRNA expression was significantly higher in subjects with T2D compared to non-diabetic control donors (Fig. 3b). Furthermore, we found that treatment of *db/db* mice with adipsin, which preserves β-cell mass in diabetic *db/db* mice[23], decreased ChREBPβ abundance in a manner

proportionate to the improved glycemia and plasma insulin levels, concordant with the idea that ChREBPβ expression contributes to the glucose toxicity seen in the *db/db* mouse model of T2D (Fig. 3c–f).

We next explored nuclear localization of ChREBPβ in human β-cells in vivo using a minimal human islet transplant model in immunosuppressed mice[24]. In this experiment, human islets were transduced with adenoviruses and then transplanted under the kidney capsules of streptozotocin-induced diabetic immunocompromised mice (Fig. 4a). 1500 islet equivalents (IEQs) were sufficient to normalize glucose levels (Fig. 4b). 500 IEQs transduced with a control Cre-expressing adenovirus was a minimal mass of β-cells sufficient to keep the animals alive, but hyperglycemic (around 400 mg/dL). When 500 IEQs were transduced with an adenovirus expressing ChREBPα, which activates the Nrf2 antioxidant pathway[8], blood glucose, plasma insulin, and glucose tolerance all approached normal levels (Fig. 4b–e). At the end of the experiment, a uninephrectomy was performed and glucose levels rose to diabetic levels, confirming that the transplanted human β-cells provided the only insulin in the recipient mice (Fig. 4b). Kidneys containing islet grafts were fixed and immunolabeled for insulin and ChREBP using the C-terminal antibody (Fig. 4f). This revealed abundant nuclear ChREBPβ in β-cells transduced with control, Cre-expressing adenovirus, but nuclear labeling was nearly absent in the ChREBP-α-treated β-cells. In addition, there was a strong correlation between glucose levels prior to the harvesting of the graft and nuclear ChREBPβ abundance (Fig. 4g). Thus, nuclear ChREBPβ expression is proportionate to glucose levels in human β-cells in vivo.

## Whereas ChREBPβ is required for adaptive β-cell proliferation and expansion of β-cell mass, depletion of ChREBPβ protects from glucolipotoxicity

To more deeply explore the physiological role of ChREBPβ in adult pancreatic β-cells, floxed ChREBPβ^lox/lox mice were crossed with MIP-Cre-ERT mice to generate a β-cell-specific tamoxifen-inducible knock out of ChREBPβ (iβKOβ)[10,25], (Fig. 5a and Supplementary Fig. 9, 10). iβKOβ mice were injected with tamoxifen or oil for 5 consecutive days (75 μg/g body weight). Two days later, they were placed on either a control chow diet or a high-fat diet (HFD) for either one or four weeks. As expected, the oil-injected mice showed nuclear ChREBP labeling with the C-terminal antibody after a high-fat diet, whereas this labeling was not observed in the tamoxifen-injected mice, confirming the high knock-out efficiency in iβKOβ mice, and supporting the notion that ChREBPβ becomes nuclear in vivo in response to high-fat feeding (Supplementary Fig. 9a). Three weeks after the first tamoxifen injection, mice on a HFD displayed a significant increase in non-fasting blood glucose levels compared to vehicle-treated littermate controls at 3 weeks (Fig. 5b). After one week, glucose tolerance was impaired on a HFD, but there was no significant difference between the oil and tamoxifen-injected groups (Fig. 5c, d). However, Ki67 immunolabeling in β-cells after one week on a HFD was significantly lower in tamoxifen-injected iβKOβ mice compared to littermate controls, demonstrating the necessity for ChREBPβ for adaptive β-cell proliferation (Fig. 5e, f). After 1 month on a HFD, the tamoxifen-injected mice became more

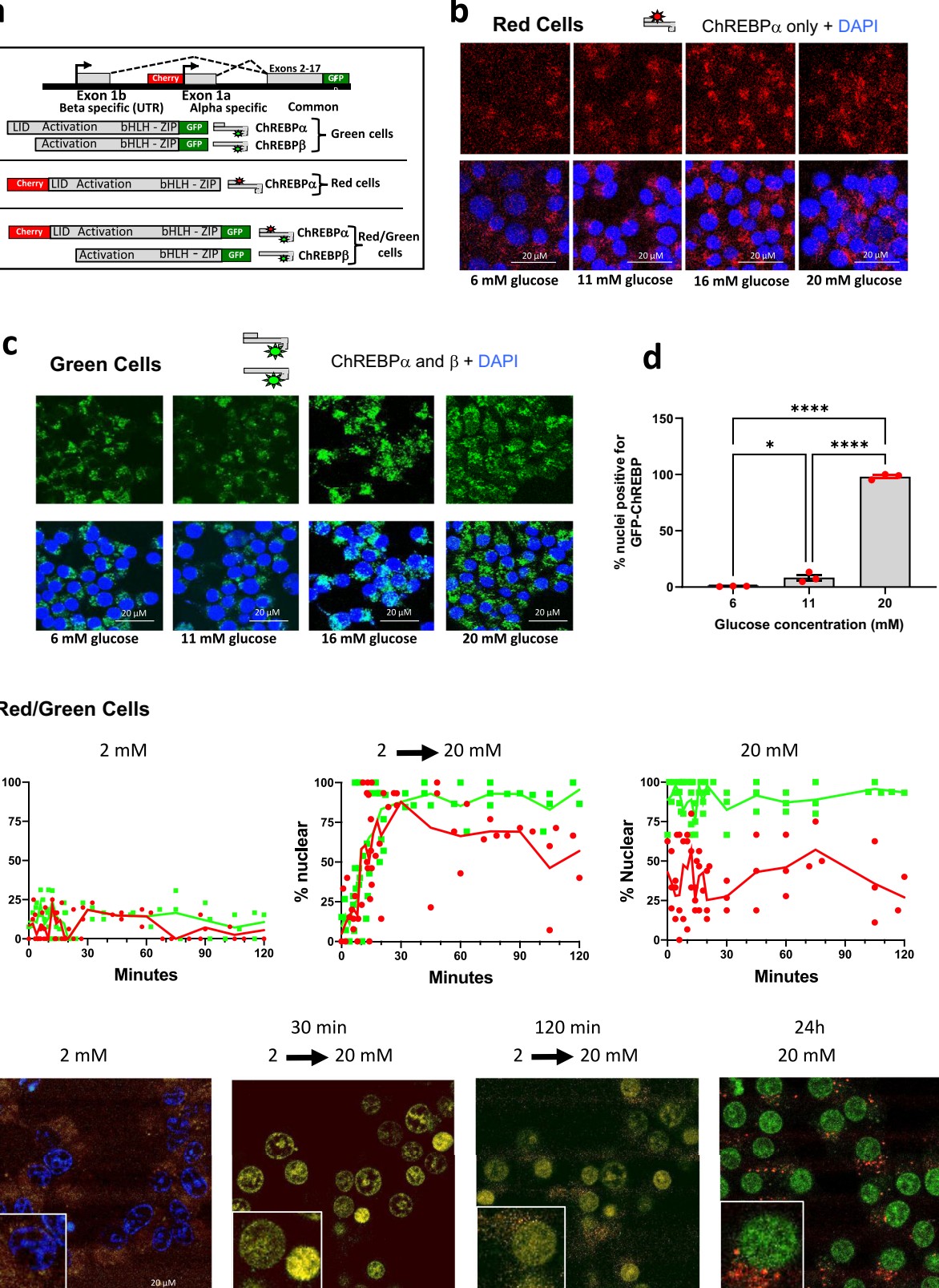

**Fig. 2 | Cellular localization of genetically edited and tagged ChREBP isoforms in response to glucose. a** Design of labeled ChREBP isoforms. mCherry and eGFP were integrated into the genome using the CRISPR/Cas9 method for gene editing in frame with exon 1a and exon 17. **b, c** Confocal live imaging was performed on Red or Green INS-1 cells after incubation with the indicated concentrations of glucose for 72 h. **d** The percent nuclear green fluorescence from **c** was determined. Data are the means ± SEM, *n* = 3, *P < 0.05; ****p < 0.0001 using one-way ANOVA. **e, f** Red/

Green INS-1 cells were incubated in 2 or 20 mM glucose or the media was changed from 2 to 20 mM glucose as indicated. Confocal live-image microscopy was performed and the percent nuclear green or red fluorescence was determined. The data presented in (**e**) represent the aggregate of 3 independent experiments. **f** Representative images from the indicated times and treatments. DAPI nuclear staining is included in the 2 mM image and excluded in the others for clarity. See also related Supplementary Figs. 4–8 and Supplementary Movie 1.

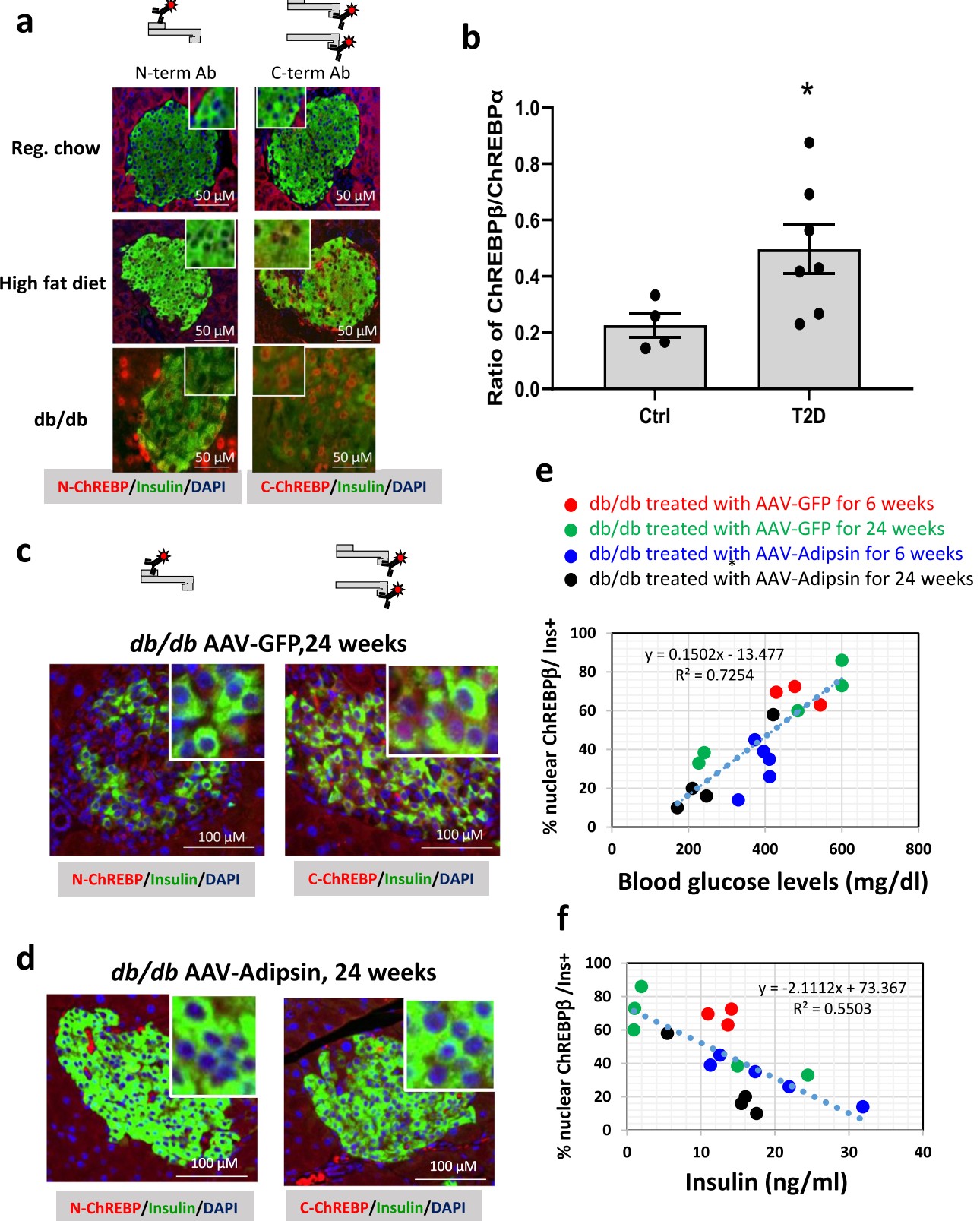

**Fig. 3 | ChREBPβ expression and nuclear localization correlates with adaptive expansion of β-cells and with glucose toxicity in diabetes. a** The N-terminal or the C-terminal antibodies recognizing ChREBP were used to stain pancreatic tissue slices from C57Bl/6 mice fed on a standard chow diet, or fed a high-fat diet for 1 week, or from *db/db* diabetic mice. All micrographs represent at least 3 independent experiments. **b** Ratio of expression of ChREBPβ to ChREBPα FPKM from RNAseq performed from FACS-sorted human β-cells isolated from non-diabetic (*n* = 4) and Type 2 diabetic subjects (*n* = 7). Each data point represents a different donor. Data are means ± SEM; *$p < 0.05$ using one-way ANOVA. **c, d** The N-terminal or C-terminal antibody recognizing ChREBP were used to immunolabel pancreatic tissue slices from diabetic *db/db* mice treated with control AAV (GFP) or with AAV expressing adipsin for 24 weeks. Correlation between percentages of nuclear ChREBPβ in insulin-positive β-cells and blood glucose levels (**e**) or blood insulin levels (**f**), where each data point represents an individual mouse. All micrographs represent at least 3 independent experiments.

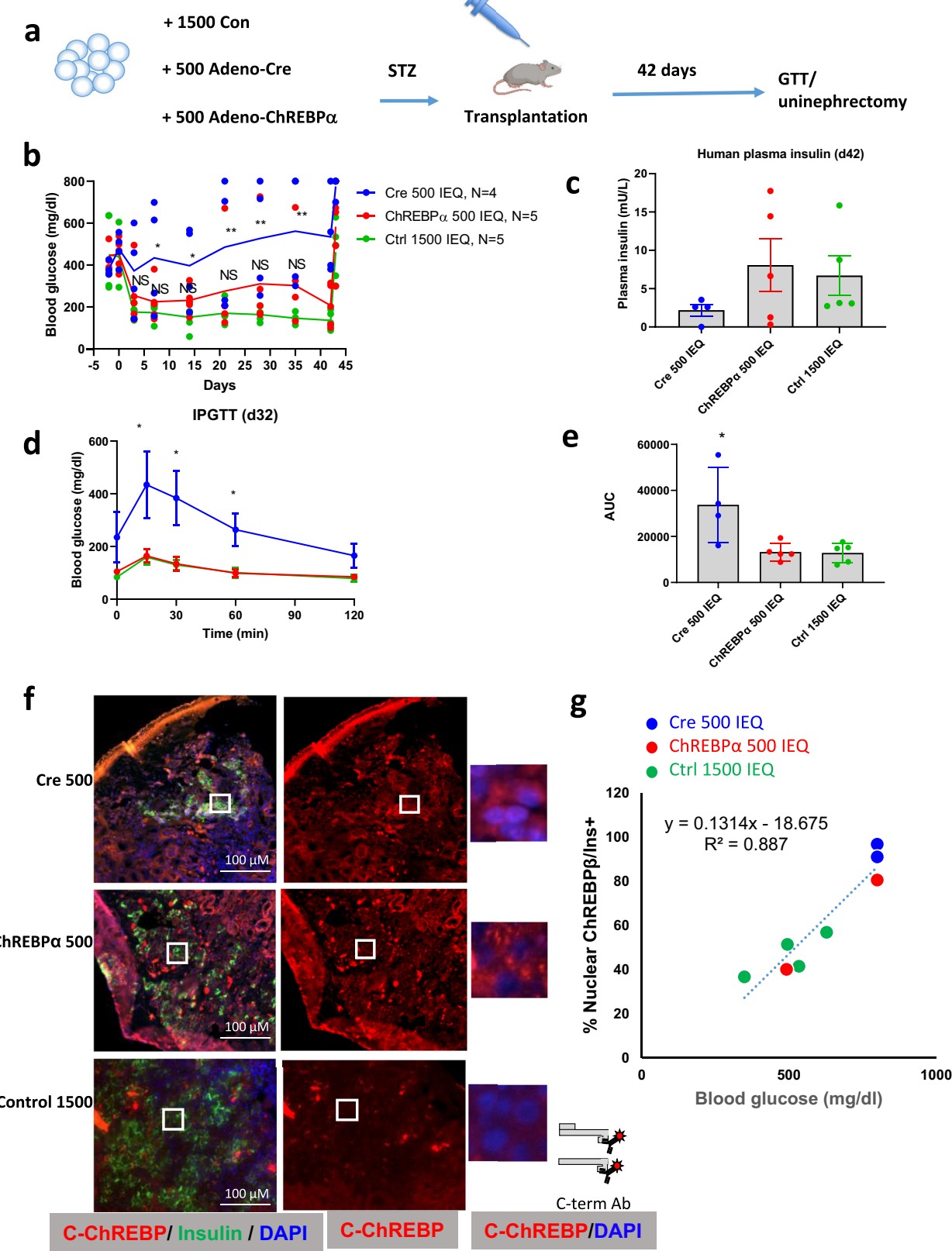

glucose intolerant compared with littermate oil-injected mice (Fig. 5g, h). Concordantly, β-cell mass was significantly lower in tamoxifen-injected iβKOβ mice fed a HFD for a month compared to iβKOβ mice injected with oil and fed a HFD, but was not different from β-cell mass in chow-fed control mice (Fig. 5i). Plasma insulin levels remained the same in mice on a HFD diet despite the increased glucose levels

(Fig. 5j). Female mice were largely protected from the effects of ChREBPβ depletion (Supplementary Fig. 10c–g). Together, these findings illustrate that ChREBPβ is necessary for adaptive β-cell mass expansion.

We repeated these experiments, crossing ChREBPβ[lox/lox] mice with INS-1-Cre[Herr] mice[26], which express Cre under control of the insulin

**Fig. 4 | ChREBPβ nuclear localization correlates with blood glucose levels in β-cells from transplanted human islets. a** Schematic of experimental design using the STZ-diabetic immunocompromised marginal islet mass model (Created with BioRender.com). Three groups of mice each received human islets from the same four to five human islet donors: Cre-transduced 500 IEQs ($n = 4$), ChREBPα-transduced 500 IEQs ($n = 5$), or 1500 untreated control IEQs ($n = 5$). The 500 IEQ groups were treated with an adenovirus expressing either Cre as a negative control or ChREBPα 24 h prior to transplantation. Glucose tolerance test was performed at day 42, a unilateral nephrectomy was performed at day 43. **b** Glucose levels were measured. **c** Circulating insulin measured using a human insulin-specific assay. **d** Intraperitoneal glucose tolerance test. **e** Area under the curve (AUC) for the three groups in (**d**). Values are means ± SEM. *$p < 0.05$; **$p < 0.01$ compared to 1500 IEQ using two-way ANOVA. **f** Representative images of insulin and C-terminal ChREBP immunohistochemistry from islet grafts of at least 3 different mice. **g** Correlation between percentages of nuclear ChREBPβ from at least 1000 insulin-positive β-cells from the grafted human islets and blood glucose levels, where each data point represents an individual transplanted mouse.

promoter at or near embryonic day 8.5, to generate an embryonic β-cell-specific knock out of ChREBP (eβKOβ), and performed the same set of experiments on two-month-old mice (Supplementary Figs. 11–13). These experiments recapitulated the results in iβKOβ mice, showing that ChREBPβ was necessary for β-cell expansion after a HFD in male (but not female) mice. Thus, ChREBPβ is dispensable for normal β-cell development under non-metabolically-stressed conditions, but is required for adaptive proliferation and expansion of β-cell mass in response to a HFD in adult male mice.

To explore underlying mechanisms, mRNA was isolated from islets of male eβKOβ mice fed one week on a chow or HFD (Supplementary Fig. 12e–g). Whereas the knockdown efficiency of ChREBPβ in the Cre-positive mice both on chow or HFD is evident and significant, ChREBPα remained unchanged between Cre-negative and Cre-positive groups, as did β-cell markers, consistent with the idea that ChREBPβ is not necessary for β-cell differentiation or maintenance of β-cell phenotype. Myc, a cell cycle regulator required for adaptive expansion of β-cell mass, and an essential factor for ChREBP activity[21,27,28], was downregulated in both chow and HFD after depletion of ChREBPβ (Supplementary Fig. 12g). This suggests that the lack of proliferation seen in Cre-positive iβKOβ and eβKOβ β-cells may be caused by a failure to induce Myc in response to the HFD. Altogether, the data from the iβKOβ and eβKOβ mice demonstrate that ChREBPβ plays a key role in adaptive β-cell proliferation but is unlikely to play a role in normal pre- and postnatal β-cell expansion during development.

### Depletion of ChREBPβ protects against glucolipotoxicity

To further explore whether ChREBPβ might play a role in glucolipotoxicity, we cultured dispersed islet cells from ChREBPβ^lox/lox mice after transduction with adenoviruses expressing either control GFP or Cre recombinase in low glucose or high glucose plus palmitate (Fig. 5k, l). Culturing islet cells in glucolipotoxic conditions led to marked increases in β-cell death as assessed using TUNEL assay (average ~50%). Strikingly, deletion of ChREBPβ completely prevented cell death. Thus, although transient increases in ChREBPβ are necessary for adaptive β-cell expansion, sustained increases in ChREBPβ are a key driver of glucolipotoxic β-cell death.

### ChREBPβ overexpression in vivo results in β-cell death and diabetes

To test if overexpressing ChREBPβ in β-cells caused β-cell death in vivo, MIP-Cre-ERT mice were crossed to mice containing a Lox-Stop-Lox Flag-tagged ChREBPβ cassette residing in the *Rosa26* locus, termed iβOEβ mice (Fig. 6a, Supplementary Fig. 14). Cre-mediated recombination resulted in the expression of flag-tagged ChREBPβ, confirmed by immunoblots and RT-PCR. Tamoxifen-mediated recombination was restricted to β-cells (Supplementary Fig.. 14b–d). Furthermore, immunostaining of pancreata displayed a marked induction of Cre in insulin-positive β-cells from the iβOEβ mice after tamoxifen treatment (Supplementary Fig. 14e).

Seven days after tamoxifen-injection, iβOEβ male mice displayed no significant change in glucose tolerance, body weight, plasma insulin or non-fasting glucose (Fig. 6b–f). However, heterozygotes and homozygotes displayed significant decreases in β-cell mass after one week (Fig. 6g). Strikingly, within 30 days after the last injection of tamoxifen, male iβOEβ mice became diabetic, evidenced by increased non-fasting blood glucose levels, impaired glucose tolerance and a marked decrease in β-cell mass, all in a gene-dose-dependent manner (Fig. 6f, h–j). Induction of ChREBPβ was associated with a significant increase in Ki67 staining, and a concomitant increase in TUNEL staining, perhaps reflecting simultaneous attempts at β-cell proliferation and cell death, or DNA damage repair activity (Fig. 6k–n). By contrast, females were partially protected, with only homozygous mice displaying significant glucose intolerance one month after tamoxifen treatment, (Supplementary Fig. 15). Thus, inducible overexpression of ChREBPβ in mice mimics glucose toxicity with β-cell destruction resulting in diabetes.

To determine whether ChREBPβ impacts β-cell function if overexpressed early in development, Lox-Stop-Lox Flag-tagged ChREBPβ mice were crossed with INS-1-Cre^Herr mice[26], to generate embryonically expressed β-cell-specific knock-in flag-tagged ChREBPβ mice, termed eβOEβ mice (Supplementary Figs. 16a–c). At 3 weeks of age, body weight was not significantly different between the Cre-negative and Cre-positive littermates, and only homozygous eβOEβ mice showed a significant increase in non-fasting or fasting blood glucose levels (Supplementary Fig. 16d, e). By 8 weeks both heterozygous and homozygous eβOBβ Cre-positive mice displayed increased non-fasting glucose levels (Supplementary Figs. 16e). At 8 weeks of age, the male mice overexpressing either one or two copies of ChREBPβ had severely impaired glucose tolerance, increased fasting and non-fasting glucose levels, and homozygous animals had severe diabetes and weight loss (Supplementary Fig. 16d–h). Concordantly, insulin levels were significantly decreased in homozygous mice, and β-cell mass was nearly absent for both heterozygous and homozygous mice (Supplementary Fig. 16i, j). Furthermore, the islet architectures of the Cre-positive mice are clearly disturbed with necrotic centers evident within islets (Supplementary Fig. 17a), with no obvious difference in somatostatin or glucagon immunolabeling, and normal alpha cell mass (Supplementary Fig. 17b). The overall graded response between heterozygotes and homozygotes demonstrated a gene dosage effect. Insulin tolerance was very similar between the Cre-negative and Cre-positive groups, indicating that the diabetic phenotype was due to catastrophic loss of β-cell mass rather than insulin intolerance in peripheral tissues (Supplementary Fig. 17c–g; see also Supplementary Movies 2 and 3). Results in female eβOEβ mice were similar to those in male mice (Supplementary Fig. 18). Thus prolonged and very high overexpression of ChREBPβ leads to β-cell death, mimicking glucotoxicity.

### Overexpression of ChREBPβ promotes a signature of increased proliferation, apoptosis, and dedifferentiation

To more deeply explore the relationship between ChREBPβ overexpression and β-cell proliferation versus death, we performed RNA-seq using INS-1 cells transduced with a control adenovirus (GFP) or an adenovirus overexpressing ChREBPβ and cultured for 48 h in 2 mM or 11 mM glucose (Fig. 7). Six differential gene expression (DGE) analyses were conducted to unbiasedly compare all possible combinations between the 4 groups. Gene Ontology (GO) terms enriched by differentially expressed genes (Supplementary Table 4) from each DGE were

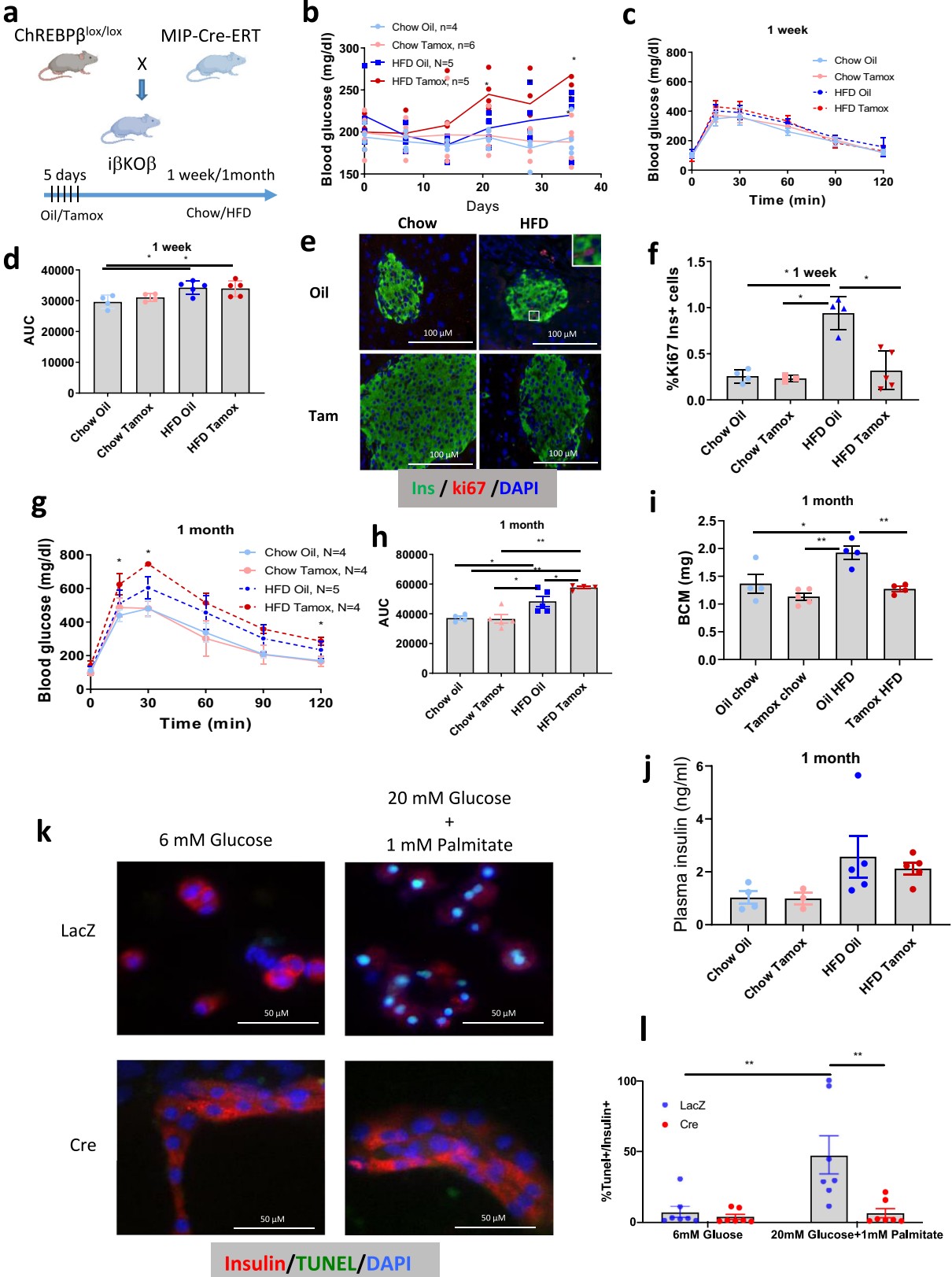

processed using the ViSEAGO R package that helps capture the biological background from the complexity of the experimental design with multiple comparisons[29]. ViSEAGO computed the semantic similarity between 471 enriched GO terms, and identified 45 clusters, and 5 superclusters of GO terms (Fig. 7a). Supercluster (1) includes clusters populated by apoptosis, cell death and proliferation clusters. Other

superclusters include transmembrane transport (2) regulation of metabolic processes (3 and 4) and cell differentiation (5). Figure 7b depicts a volcano plot comparing GFP and ChREBPβ-treated INS-1 cells cultured in 11 mM glucose. GO pathway analysis revealed that the top two pathways associated with ChREBPβ overexpression in 11 mM glucose were proliferation and apoptosis (Fig. 7c). Nearly every apoptosis

**Fig. 5 | ChREBPβ is required for adaptive β-cell proliferation and expansion of β-cell mass, and loss of ChREBPβ prevents glucolipotoxicity. a** Schematic showing generation of β-cell specific, inducible ChREBPβ knockout mice (iβKOβ, created with BioRender.com). **b** Blood glucose levels after the indicated treatments and times. **c, d** Glucose tolerance test and area under the curve measurements after 1 week on a chow or high-fat diet (HFD). **e, f** Percent Ki67-positive/insulin-positive cells in pancreata from iβKOβ mice after 1 week of HFD. **g, h** Glucose tolerance test and area under the curve measurements after 1 month on a chow or high-fat diet (HFD). **i, j** β-cell mass and Plasma insulin were measured after 1 month on a chow or high-fat diet (HFD). Data are means ± SEM; $N = 4$ or 5 as indicated; *$p < 0.05$; **$p < 0.01$ using two-way ANOVA. **k** Islets were isolated from floxed ChREBPβ mice, dispersed and transduced with adenoviruses expressing either GFP or Cre recombinase. Cells were cultured as indicated and subjected to a TUNEL assay after 48 h and immunostained for insulin and DAPI. **l** Percentage of insulin-positive/TUNEL-positive cells. Data are means ± SEM; $N = 7$; *$p < 0.05$; **$p < 0.01$ using two-way ANOVA.

marker was upregulated by ChREBPβ, both in 2 mM and 11 mM glucose (Fig. 7d). In addition, *Txnip*, which drives β-cell glucose toxicity and is a major target gene of ChREBP[30], was highly upregulated by ChREBPβ in both 2 mM and 11 mM glucose (Fig. 7e). Key cell cycle regulator genes were generally induced by increased glucose, and overexpression of ChREBPβ led to an even greater increase in their expression (Fig. 7f). By contrast, β-cell identity markers were markedly decreased by ChREBPβ overexpression (Fig. 7g). Thus, overexpression of ChREBPβ behaves much like overexpression of Myc, with a signature supporting both proliferation and apoptosis in β-cells, and with a general effect of decreasing β-cell identity while increasing the transcription of most of the genes examined[31,32].

### Rescue from ChREBPβ-mediated β-cell death by ChREBPα and Nrf2

Since overexpression of ChREBPα does not result in β-cell death, but rather augments glucose-stimulated β-cell proliferation via activation of the antioxidant Nrf2 pathway[5,8], we queried whether co-expression of ChREBPα with ChREBPβ could rescue β-cells from ChREBPβ-mediated cell death. We first tested the hypothesis that the ratio of ChREBPα:ChREBPβ is an important determinant of β-cell apoptosis. INS-1 cells were transduced with a constant MOI (150) of adenoviruses expressing either ChREBPα or ChREBPβ. Increasing MOIs of each virus were compared, using LacZ-expressing adenovirus to maintain a constant viral load (Fig. 8a). Cell death, as measured by Annexin V staining and flow cytometry, was evident when the ratio of ChREBPβ to ChREBPα was greater than one. In addition, titration of ChREBPα into ChREBPβ-expressing cells reduced β-cell death. We next isolated islets from Lox-stop-Lox ChREBPβ mice, and induced ChREBPβ using a Cre adenovirus in the absence or presence of an adenovirus expressing ChREBPα, and then measured β-cell death after 48 h by counting TUNEL- and insulin-positive cells. There was almost universal apoptosis in cells overexpressing ChREBPβ, but co-expression of ChREBPα with ChREBPβ rescued β-cell death (Fig. 8b, c). Overexpression of ChREBPα activates the antioxidant Nrf2 pathway[8]. Thus, we explored whether CDDO-Me, an Nrf2 activator, might also rescue β-cells from isolated ChREBPβ-mediated cell death using islets from Lox-stop-Lox ChREBPβ mice. This proved to be true (Fig. 8b, c). Thus, overexpression of ChREBPα or activation of Nrf2 rescues murine β-cells from the cytotoxicity of ChREBPβ overexpression.

We asked if ChREBPα could rescue ChREBPβ-mediated β-cell death in human β-cells. ChREBPβ was overexpressed using an adenovirus in dispersed human islets. Overexpression of ChREBPβ resulted in pronounced β-cell apoptosis, as assessed by insulin immunolabeling and TUNEL assay (Fig. 8d, e). We also immunolabeled the same cells for Ki67 and observed a marked Ki67 labeling in β-cells, likely reflecting DNA damage in dying β-cells [Fig. 8f; refs. 2,33]. Overexpression of ChREBPα had no effect on either proliferation or cell death. However, co-expression of ChREBPβ and ChREBPα resulted in a complete absence of cell death, but retention of robust β-cell proliferation. In separate experiments, ChREBPβ-transduced human islet cells were treated with CDDO-Me, an Nrf2 activator (Fig. 8e, f). CDDO-Me reduced ChREBPβ-mediated TUNEL staining and induced or permitted robust Ki67 immunolabeling, strongly suggesting that the rescue effect of ChREBPα was at least in part due to activation of the Nrf2 antioxidant pathway.

## Discussion

β-cells have the remarkable ability to adapt and compensate for increased demand for insulin by expanding β-cell mass, as happens in response to a hypercaloric Western diet in rodents or pregnancy in rodents and humans[2,34]. Diabetes results from an inability of β-cells to compensate for increased demand for insulin, or from decompensation resulting from a glucotoxic environment from prolonged hyperglycemia[1,35,36]. Here we find that ChREBPβ is a key modulator of both compensation and decompensation of β-cells (Fig. 9). Major findings include: (1) in response to increased glucose, ChREBPα rapidly and transiently migrates to the nucleus, and through the resulting positive feedback induction, ChREBPβ becomes the major nuclear form of ChREBP. The nuclear abundance of ChREBPβ correlates with the degree of hyperglycemia in rodent and human β-cells. Thus, nuclear ChREBPβ is a biomarker for the β-cell response to hyperglycemia. (2) The induction of ChREBPβ is required for the normal early physiological adaptive expansion of β-cells in response to a high-fat diet. (3) Chronic ChREBPβ overexpression results in β-cell death and diabetes. (4) Deletion of ChREBP-β prevents glucolipotoxicity in cultured mouse β-cells. (5) ChREBP-β-mediated glucose toxicity can be mitigated by exogenous expression of ChREBPα, or with activation of the Nrf2 antioxidant pathway.

The canonical view of ChREBP translocation posits that ChREBP remains in the cytoplasm under low glucose conditions, and then rapidly translocates to the nucleus in response to the metabolism of glucose[19]. Most assays that characterize ChREBP translocation have been performed either with antibodies that do not distinguish between ChREBPα and ChREBPβ, or by ectopic expression of epitope-tagged ChREBPα[17,18,37]. Here we used tools designed to distinguish the two major endogenous isoforms and found that ChREBPα only transiently migrates to the nucleus in response to glucose and then induces the production of ChREBPβ. Importantly, ChREBPβ is generated from a powerful positive feedback loop driven by ChoREs near an alternative transcription start site. Alternative splicing replaces exon 1a with exon 1b, leading to a mRNA encoding ChREBPβ that lacks the LID domain and an NES, and so is more transcriptionally potent and more likely to be nuclear than ChREBPα[11]. Thus, with sustained, prolonged glucose metabolism, ChREBPβ becomes the predominant ChREBP isoform bound to DNA. The functional interaction between the two isoforms of ChREBP provides the molecular machinery for acute glucose regulation by ChREBPα, and the regulation of adaptive responses for longer exposure to glucose by ChREBPβ, and the potential for pathological consequences after chronic and unrestrained positive feedback expression of ChREBPβ.

While transcriptional regulation and localization of the two splice isoforms, as demonstrated in this study seem to be of high importance for the regulation of ChREBP and activity, there are additional mechanisms that have been previously shown to tightly regulate ChREBP location and activity. ChREBP is regulated by carbohydrate metabolites and other metabolic signals[19,38], including Ca++ flux, which dissociates its binding to sorcin allowing nuclear translocation in beta cells[18], and several posttranslational modifications [reviewed in[39]], which may affect its stability and the binding of co-factors and co-activators[40,41]. Additionally, nuclear retention of ChREBP's mRNA[42], and sequestration of ChREBP's heterodimer partner, Mlx, in lipid droplets play important roles in regulating ChREBP activity[43]. Furthermore,

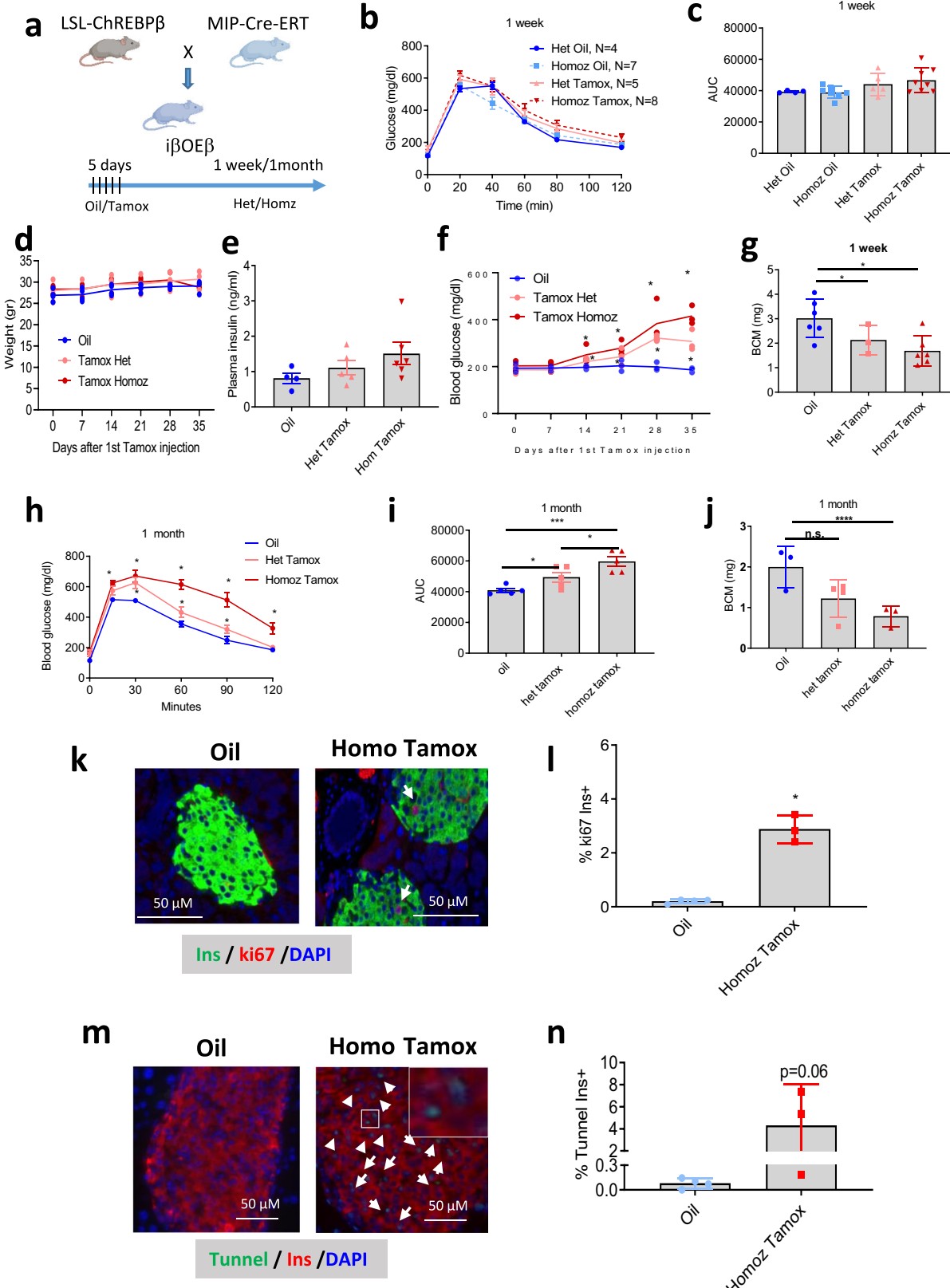

there are other mechanisms that have not been fully explored that may regulate the ratio of ChREBP isoform activity during increasing metabolic stress including differential mRNA and protein stability and differential protein translation rates. Clearly, a transcription factor that plays such an important role in beta cell function, proliferation, and apoptosis should be tightly controlled.

We have previously shown that the induction of ChREBPβ is necessary for a full transcriptional and proliferative response to glucose in rodent β-cells in vitro[10]. Here we show that ChREBPβ is necessary for adaptive β-cell proliferation and expansion of β-cell mass using models that remove ChREBPβ either conditionally in adults, or embryonically. One important observation is that removal

**Fig. 6 | Overexpression of ChREBPβ leads to β-cell death, glucose intolerance and diabetes. a** LSL-ChREBPβ mice were bred with MIP-Cre-ERT mice to generate inducible β-cell-specific overexpressing ChREBPβ mice, termed iβOEβ (created with BioRender.com; see also Supplementary Figs 16 and 17). Presented are measurements from male mice (see also Supplementary Fig. 18 for female mice). **b, c** Glucose tolerance test and area under the curve (AUC) one week after vehicle oil or tamoxifen treatment. **d** Weekly measures of whole body weight. **e** Plasma Insulin levels after one week. **f** Non-fasting blood glucose levels for the indicated times and treatment groups. **g** β-cell mass (BCM) one week after vehicle oil or

tamoxifen treatment. **h, i** Glucose tolerance test and area under the curve (AUC) one month after vehicle oil or tamoxifen treatment. **j** β-cell mass (BCM) one month after vehicle oil or tamoxifen treatment. Data are means ± SEM, $N = 4–6$; *$p < 0.05$; **$P < 0.001$; ****$P < 0.0001$ using two-way ANOVA. **k, l** Percent Ki67-postive and insulin-positive cells in pancreata from male homozygous iβOEβ mice 1 week after oil or tamoxifen injection. **m, n** Percent TUNEL-positive and insulin-positive cells in pancreata from male homozygous iβOEβ mice 1 week after oil or tamoxifen treatment. Data are means ± SEM from five mice/group. *$p < 0.05$ using two-way ANOVA.

of ChREBPβ did not affect glucose homeostasis unless the mouse was challenged with a HFD. Thus, ChREBPβ is dispensable for normal β-cell development and β-cell function under non-stressed circumstances. This is consistent with observations of very low expression levels of ChREBPβ mRNA in β-cells cultured in low glucose and with global ChREBP knock out mice, which display relatively normal β-cell function[10,44]. Another striking observation was that deletion of ChREBPβ protected β-cells from glucolipotoxic cell death. Since ChREBP is lipogenic, and inhibition of lipogenesis promotes fatty acid oxidation and protection from ceramide-mediated β-cell death[19,45,46], a reasonable explanation for the protective effect of ChREBPβ depletion in β-cells may be decreased lipogenesis and an increased ability to process the exogenous palmitate.

Whereas overexpression of ChREBPα does not cause cell death[5,8], overexpression of ChREBPβ resulted in robust β-cell apoptosis, resulting in decreased β-cell mass and diabetes. As with our studies, Chan and colleagues found a similar result in INS-1 cells overexpressing a constitutively active version of ChREBP that lacks the LID domain and thus is functionally and structurally similar to ChREBPβ[14]. That ChREBPβ drives apoptosis is consistent with the fact that *Txnip* is a target gene of ChREBP and that depleting β-cells of ChREBP or Txnip prevents glucotoxic mediated cell death[7,14,47]. It seems that the ratio of ChREBPβ to ChREBPα is critical for β-cell fate, and that a threshold of ChREBPβ must be reached for pathological results to occur. We observed a dose- and time-dependent component of ChREBPβ-mediated β-cell death in vivo, FACs-sorted human β-cells from T2D donors displayed a higher ChREBPβ:ChREBPα ratio than from non-diabetic donors, and increasing the ChREBPα abundance mitigated ChREBP-β-mediated cell death in vitro and in vitro in both rodent and human β-cells. These results are in concert with studies showing high levels of ChREBP correlate with poor diagnosis and increased rates of proliferation in certain forms of cancer[48]. Indeed, in many respects, ChREBPβ behaves very much like Myc in β-cells. Myc is induced by glucose and its overexpression is closely correlated to glucose toxicity and the reduction of β-cell-enriched genes, including insulin itself[49]. Furthermore, overexpression of ChREBPβ, much like Myc, results in a general amplification of actively transcribed genes[31,50]. Collectively, these results suggest that ChREBPα and ChREBPβ drive different transcriptional programs. On the one hand, overexpression of both ChREBPα and ChREBPβ result in decreased expression of β-cell markers [ref. 51 and this study]. On the other hand, overexpression of ChREBPβ leads to hyper-expression of Txnip and cell death, whereas overexpression of ChREBPα activates the Nrf2 antioxidant pathway via unknown mechanism, resulting in enhanced glucose-stimulated proliferation and protection from ChREBPβ-mediated cell death [ref. 8 and this study]. The identity of the different gene targets of the two ChREBP isoforms and the mechanisms by which they are regulated are areas of active investigation.

Interestingly, Shalev and colleagues demonstrated that ChREBPβ overexpression decreases ChREBPα abundance and postulated that a role of ChREBPβ is to prevent ChREBPα from mediating glucose toxicity[13]. Those experiments were performed at relatively early time points in vitro and did not assess apoptosis. We suspect that had

longer term experiments in the studies of Jing et al. would have revealed results similar to those reported here.

We found it possible to mitigate the effect of ChREBPβ by either restoring the balanced ratio of ChREBPβ to ChREBPα, or by activating the Nrf2 antioxidant pathway. The latter effect follows from our previous study demonstrating that overexpression of ChREBPα results in the activation of Nrf2[8]. Considering that ChREBPβ drives glucose toxicity, it is not surprising that activation of an antioxidant pathway alleviates its pathological effects. Glucose toxicity can be alleviated and β-cell function improved by a large variety of antioxidants, particularly those that activate the Nrf2 pathway[15]. Of note is the study by Kjørholt et al. demonstrating that hyperglycemia rather than hyperlipidemia leads to β-cell dedifferentiation and decreased insulin secretion in *db/db* mice, and a number of studies demonstrating that normalizing blood glucose levels with insulin or SGLT2 deletion improves β-cell function[52–54]. We suggest that these effects are likely due, in part, to cessation of the positive feedback loop producing ChREBPβ and a restoration of the ChREBPα:ChREBPβ ratio and redox balance.

A number of Nrf2 activators have been or are currently being tested in clinical trials for the treatment of diabetes. Bardoxolone methyl (CDDO-Me) was investigated for treatment of chronic kidney disease in people with diabetes, but this clinical trial was terminated early due to cardiovascular safety concerns[55]. A new Phase II clinical trial with CDDO-Me has recently started, excluding at-risk patients[56]. Several natural compounds that activate Nrf2 are also under clinical trials, though many have anti-inflammatory effects acting through pathways other than Nrf2, and point out the very real possibility of beneficial off target side effects[57]. Clearly more studies are needed to improve tissue and target specificity of Nrf2 activator compounds before realizing their potential for increasing or preserving β-cell mass in diabetes.

We acknowledge that this study was limited to β-cells. It will be important to determine whether the mechanistic details of the positive feedback loop as described here apply to other tissues affected by glucose toxicity such as kidney, liver and adipose tissue. In addition, we did not define the exact mechanism of ChREBPβ-mediated β-cell death. It will be important to understand how ChREBP-β integrates with other mechanisms of glucose and lipid toxicity including ER stress, inflammasome activity, mitochondrial dysfunction and dysregulation of Foxo1[58–61].

In summary, we have clarified the role of ChREBP in β-cells by examining the unique functions of the two major forms of ChREBP, ChREBPα and ChREBPβ. Developing tools that distinguish the two isoforms has allowed a precise delineation of the molecular mechanism by which glucose initiates a positive feedback loop that allows ChREBPβ to become the major nuclear isoform of ChREBP in the context of adaptive responses such as from a HFD. Loss-of-function studies revealed that ChREBPβ is necessary for adaptive expansion. However, excessive ChREBPβ, as may happen from hyper-activation of the positive feedback loop with prolonged hyperglycemia or diabetes, results in β-cell apoptosis. Mitigation of ChREBPβ toxicity is possible by addition of ChREBPα to restore ChREBPα:ChREBPβ ratio, or by activation of the Nrf2 antioxidant pathway.

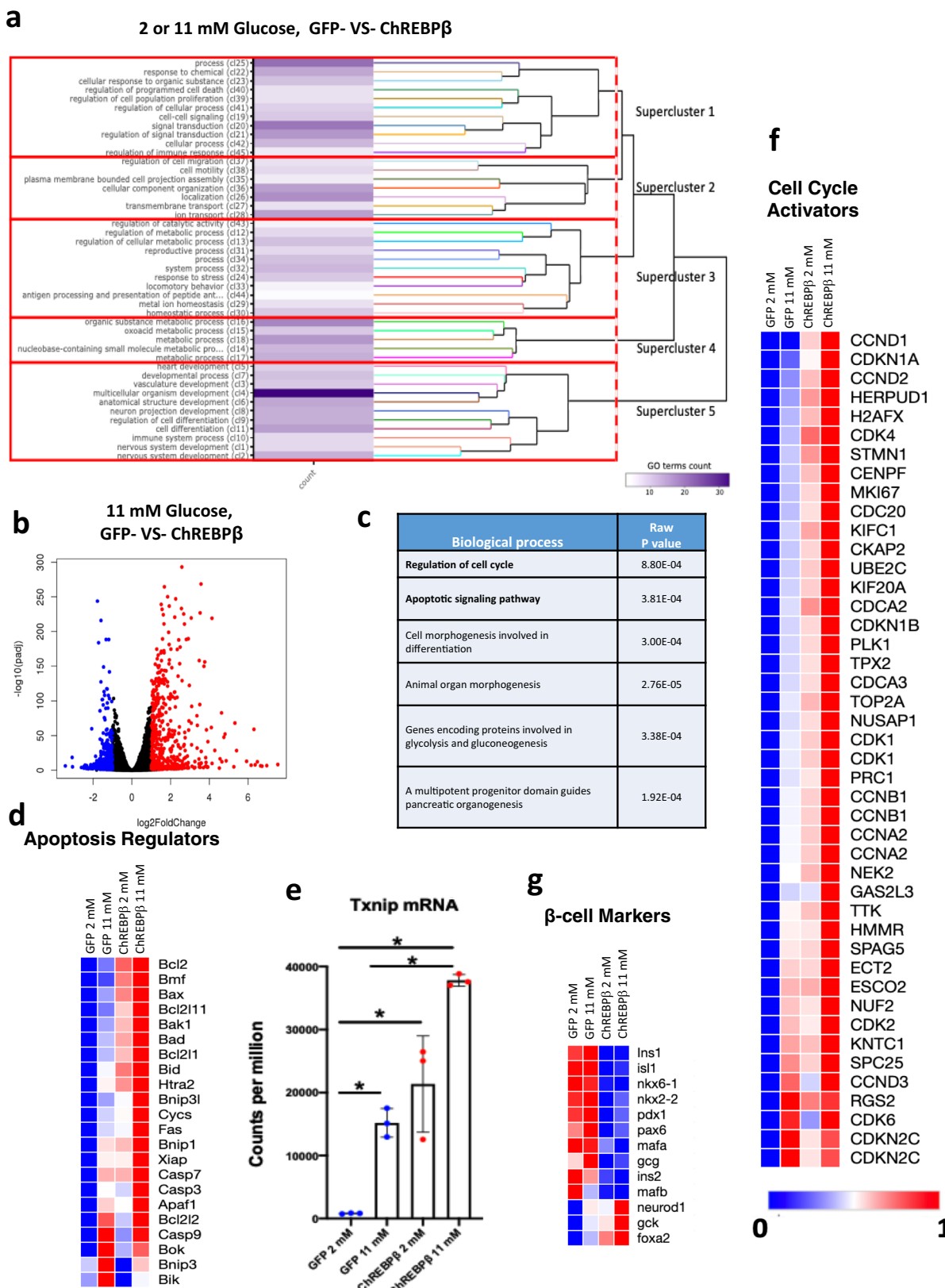

**Fig. 7 | ChREBPβ initiates programs of cell cycle regulation and apoptosis when overexpressed in Ins-1 cells.** Ins-1 cells were transduced with an adenovirus expressing GFP or ChREBPβ and cultured for 48 h in 2 or 11 mM glucose. RNA was collected and RNA-seq was performed. **a** Heatmap of the cluster of GO terms. The 471 GO terms (pathways) enriched by differentially expressed genes across the 6 DGEs from the 4 groups (see text) are grouped into 45 clusters of GO terms. The hierarchical tree scoring the distances between each of the 45 clusters originates superclusters that are boxed in red. The superclusters highlight the main cellular functions common to the different DGEs. **b**, **c** Volcano plot and gene ontology of biological processes affected by ChREBPβ compared to GFP-treated INS-1 cells cultured in 11 mM glucose. **d** Heat maps of the average expression levels of apoptosis regulators **e** *Txnip* mRNA expression; means of 3 independent measures, error bars are SEM; *$p < 0.05$. **f**, **g** Heat maps of the averages of differentially expressed cell cycle activators or β-cell marker genes. Values represent the average of three independent experiments and genes are sorted by the Pierson correlation of nearest neighbors.

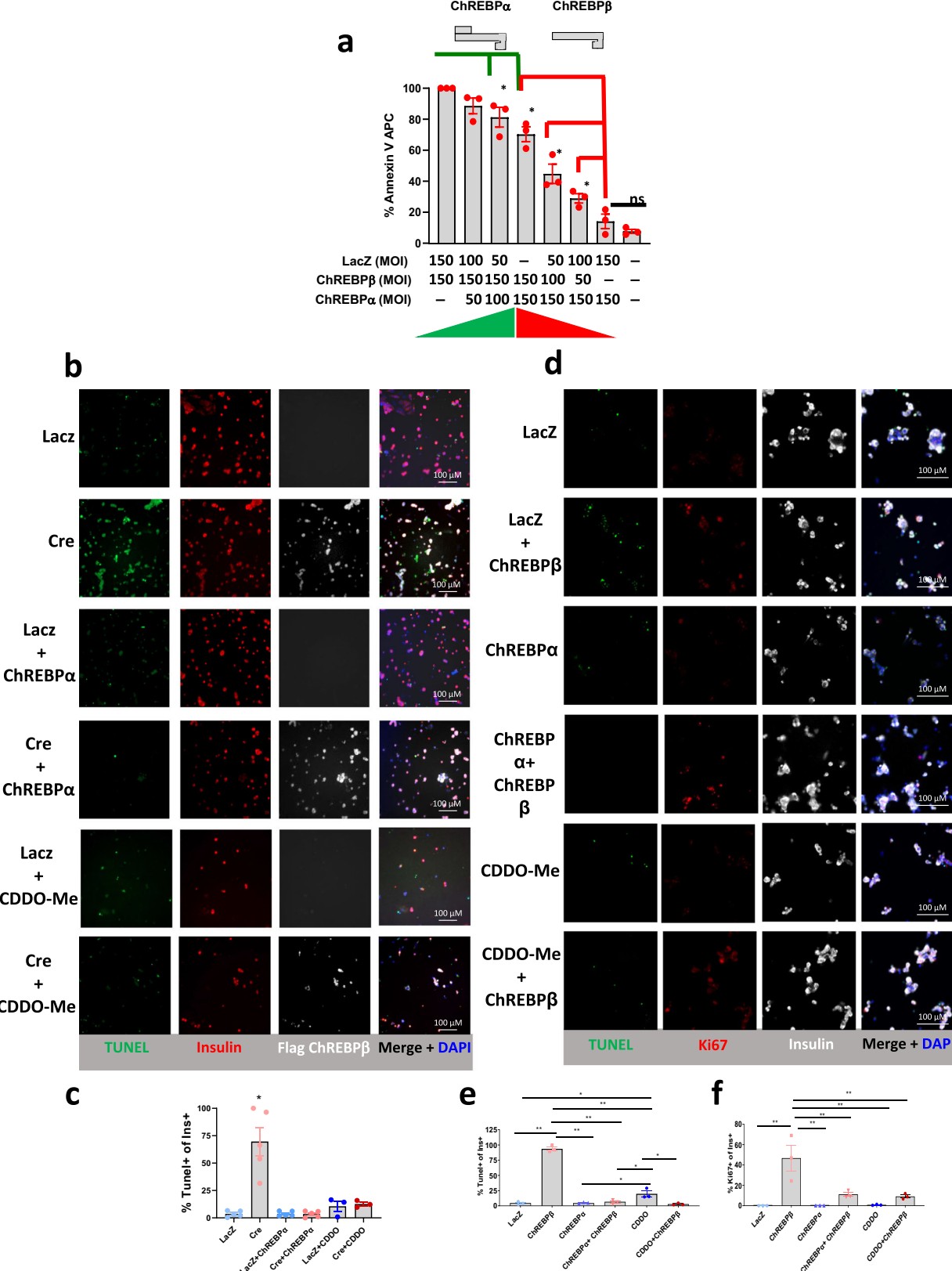

**Fig. 8 | ChREBPα and activation of Nrf2 mitigates ChREBPβ-mediated β-cell death. a** The indicated amounts of ChREBPα and ChREBPβ were transduced into INS-1-derived 832/13 cells and 48 h later apoptosis was measured with Annexin V staining. Error bars are SEM, $N = 3$, $*p < 0.05$. **b** Islets were isolated from Lox-Stop-Lox ChREBPβ mice, dispersed and transduced with the indicated adenovirus or 10 μM CDDO-Me for 48 h. Cells were immunostained or processed for TUNEL assay

as indicated. **c** Percent of TUNEL+/insulin+ cells from (**b**). **d** Human islets were dispersed and transduced with the indicated adenovirus or 10 μM CDDO-Me for 48 h and immunostained as indicated. **e** Percent of TUNEL+/insulin+ cells from (**d**). **f** Percent of Ki67+/insulin+ cells from (**d**). Data are the means ± SEM, $n = 3–4$, $*p < 0.05$, $**p < 0.01$ using one-way ANOVA; ns not significant.

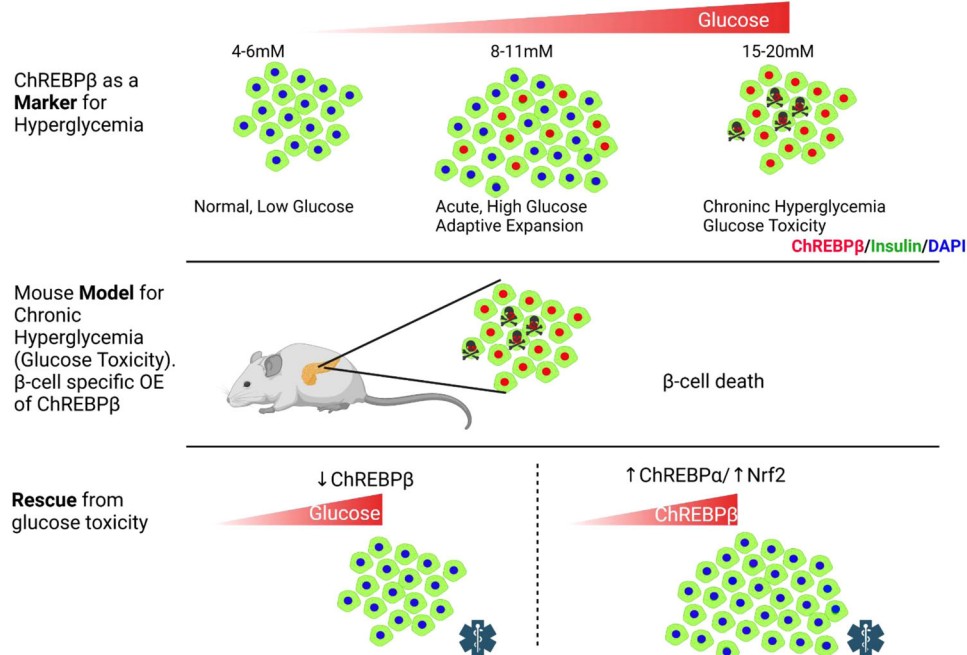

**Fig. 9 | ChREBPβ is required for adaptive β-cell expansion but contributes to glucose toxicity with prolonged hyperglycemia and effects of its over-expression can be rescued by activation of the Nrf2 antioxidant pathway.** Nuclear expression of ChREBPβ increases in β-cells with increased glucose concentrations. Loss of function experiments demonstrate that ChREBPβ is necessary for adaptive expansion of β-cells after a high fat diet. Gain of function experiments demonstrate that ChREBPβ overexpression mimics glucose toxicity. Deletion of ChREBPβ protects β-cells from glucolipotoxicity and ChREBPβ-mediated β-cell death can be mitigated by overexpression of ChREBPα or by activation of the Nrf2 antioxidant pathway. Created with BioRender.com.

## Methods

### Cell Culture
INS-1-derived 832/13 cells (a kind gift form Dr. Chris Newgard, Duke University)[62] were cultured as described previously[21]. Mouse islets were isolated by Histopaque gradient following collagenase P injection to the pancreatic duct, as previously described[63]. A day following the isolation islets were dispersed using 0.05% Trypsin. Islet cells were maintained in RPMI containing 5.5 mM glucose, 100 U/mL penicillin, 100 mg/mL streptomycin and 10% FCS as previously described[64]. Human beta cells were isolated from human cadaveric islets donors provided by the NIH/NIDDK-supported Integrated Islet Distribution Program (IIDP) (https://iidp.coh.org/overview.aspx), and from Prodo Labs (https://prodolabs.com/), the University of Miami, the University of Minnesota, the University of Wisconsin, the Southern California Islet Cell Resource enter, and the University of Edmonton, as summarized in Supplementary Table 3. Informed consent was obtained by the Organ Procurement Organization (OPO), and all donor information was de-identified in accord with Institutional Review Board procedures at The Icahn School of Medicine at Mount Sinai (ISMMS). Human islets were cultured and dispersed as previously described[5].

### Adenovirus
Ins-1 cells, mouse islets or human islets were transduced with adenovirus at multiplicity of infection (MOI) of 150, unless otherwise indicated. Cells were seeded in RPMI with 100 U/mL penicillin, 100 mg/mL streptomycin in the presence of the virus for 16 h. The following day, FBS was added to a final concentration of 10% and glucose concentration was adjusted to the concentration of the treatment. Cells were harvested or fixed 72 h after transduction and 56 h in the noted glucose concentrations. Flag-tagged mouse ChREBPα and mCherry tagged mouse ChREBPβ were cloned into pDEST and adenoviruses were generated in HEK293 cells (ATTC cat# CRL-3216) as previously described[65]; ChREBP cDNAs were kind gift from Dr. Howard Towle (U. of Minnesota). The Nrf2 adenovirus was previously described in[8].

### Immunofluorescence
After islet dispersal with 0.05% trypsin, cells were plated on 12-mm laminin coated glass coverslips placed in 24-well plates[24,66]. Islet cells were either uninfected or transduced and cultured as described above at a MOI of 150 of the adenoviruses indicated. Cells were then rinsed with PBS and fixed in 4% paraformaldehyde, and β-cell proliferation was determined by immunolabeling for Ki67 and Insulin. Confocal images were acquired using the Zeiss LSM 880 Airyscan. For time course experiments, cells were in a temperature and $CO_2$ controlled chamber throughout the experiment. Images were taken at 10 min intervals for the first 2 h following by hourly acquisition for a total of 10 h. Nuclear quantifications were performed by counting the overlap of DAPI positive nuclei and the marked fluorophore. Approximately 150 cells were counted for every time point.

### Antibodies used
Antibodies used were: anti-insulin guinea pig polyclonal at 1:000 from DAKO (cat# A0564, AB_10013624; anti-glucagon mouse monoclonal at 1:500 from Abcam (cat# ab10988, AB_297642); anti-ChREBP C-term rabbit polyclonal at 1:250 from Novus (cat # NB400-135, AB_10002435); ChREBP N-term rabbit polyclonal at 1:250 generated by Genscript and validated in Supplementary Fig. 3; anti-Flag rabbit polyclonal at 1:500 from Cell Signaling (cat# 2368S, AB_2217020); Anti-green fluorescent protein (GFP) chicken polyclonal at 1:100 from Aves-Labs (cat# GFP-1020, AB_10000240); anti-Cherry (red fluorescent protein) rabbit polyclonal at 1:1000 from Rockland (cat# 600401379S, AB_11182807); anti-Ki67 rabbit monoclonal at 1:250 from ThermoScientific (cat# MA5-14520, AB_10979488), see Supplementary Table 2.

## Genetically modified mice

Mice with β-Cell-specific inducible knockout or knockin of ChREBPβ were generated by combining either MIP-CreERT mice[25] or RIP-Cre[Herr] mice [ref. 26, MMRC-Jackson Labs] with floxed ChREBPβ mice[10] or with 3xFlag-tagged ChREBPβ transgenic mice, respectively. To generate 3xFlag-tagged ChREBPβ transgenic mice, a murine 3xFlag-tagged ChREBPβ transgene was subcloned into a modified Rosa26-pCAG-LSL-WPRE-bGHpA targeting vector between the LSL and WPRE sequences[67]. The LSL sequence contains loxP- Stop codons - 3x SV40 polyA – loxP. The targeting vector was linearized and transfected into the 129/B6 F1 ES cell line G4. G418-resistant ES clones were screened by PCR. Positive ES clones were injected into C57BL/6J blastocysts in the Boston Nutrition Obesity Research Center Transgenic Core to obtain chimeric mice following standard procedures. Chimeric mice were bred with C57BL/6J mice to obtain germline transmitted F1 mice and these mice were backcrossed onto the C57BL/6J background for more than 10 generations. All mice used in this study were in a C57BL/6J mouse background. Mice were periodically outbred to C57Bl/6 to avoid off target effects and to minimize growth hormone effects caused by breeding the Cre allele to homozygosity, as well as to obtain Cre-negative littermate controls[68]. Cre-mediated recombination was achieved by intraperitoneal injection for 5 consecutive days of 75 μg/g body weight of tamoxifen (Tam) (Sigma-Aldrich) dissolved in corn oil. All protocols were performed with the approval of and in accordance with guidelines established by the Icahn School of Medicine at Mount Sinai Institutional Animal Care and Use Committee.

## Islet transplantation

Human islets were transplanted into the renal subcapsular space as described previously[24,66,69]. Numbers of human islet equivalents are described in the Figure Legends.

## HFD feeding

14 week-old mice were fed with a lard-based HFD (41% kcal from fat) (TD 96001; Harlan Teklad) or a regular diet (RD) (13.1% kcal from fat) (Purina PicoLab 5053; LabDiet). After 7 or 30 days, body weights, non-fasting blood glucose, and plasma insulin were measured and pancreata harvested and processed for histological studies or islet isolation.

## Glucose homeostasis

Blood glucose was determined by glucometer (AlphaTrack 2) and plasma insulin by ELISA (Mercodia). An intraperitoneal glucose tolerance test was performed in mice fasted for 16–18 h and injected intraperitoneally with 2 g d-glucose/kg. Insulin tolerance test was performed in non-fasted mice IP-injected with human insulin (1.5 units/kg). Glucose-stimulated insulin secretion and insulin measures were performed as previously described[24,70].

## Immunohistochemistry and analysis of β-cell proliferation and mass

Paraffin-embedded pancreas sections were immunolabeled with antibodies for insulin (Dako) and Ki67 (Thermo Fisher Scientific), and at least 2000 β-cells were blindly counted per mouse. β-cell mass was measured from at least three insulin-labeled pancreas sections per mouse, at least 5 μm from each other, and quantified using ImageJ (National Institutes of Health). For 3D pancreatic imaging, tissue was cleared using the iDISCO method for tissue clearing as previously described[71]. Z-stack images of insulin-positive pancreata were acquired using Ultramicroscope II (LightSheet) 3D reconstitution of the images and quantification of beta cell mass using Imaris (9.7.2) software.

## Generation of CRISPR/Cas9 INS-1 cells

INS-1 cells were transfected with Lipofectamine 2000 (Invitrogen) according to the manufacturer's recommendations using two

plasmids: the first was the pX330 (Addgene) to which the PAM containing oligos were ligated into Bbs1 sites using T4 ligase (Promega, overnight according to manufacturer's instructions). For tagging the N-terminus of ChREBP, the guide RNA was constructed using the following primers: forward CACCGGTCGTCCCCAGCCCGGATTCGG and reverse AAACCCGAATCCGGGCTGGGGACGACC. For tagging the C-terminus of ChREBP, the guide RNA was constructed using the following primers: forward CACCGGACACGTCCCTCTCGATCCTGG and reverse AAACCCAGGATCGAGAGGGACGTGTCC. The first plasmid was co-transfected along with a second plasmid (pUC57) containing homology arms and the sequence for mCherry or eGFP, which was generated using GeneScript gene synthesis (sequence available upon request). Cells were allowed to recover after transfection and were expanded for 7–10 days after which they were collected by fluorescence-activated cytometric sorting at the Mount Sinai Flow Cytometry Core to isolate the fluorescent population of cells. The sorted population of cells was transfected as before, and the cycle of transfection and sorting was repeated 2–3 times to obtain a homogenous population of INS-1 cells expressing the desired fluorescent protein fused to endogenous ChREBP.

## TUNEL and Annexin V assays

TUNEL labeling was performed according to the manufacturer, using the DeadEnd Fluormetric TUNEL System (Cat#G3250, Promega). Annexin V and PI staining were performed as described previously[72] to distinguish dead (PI-AnnexinV+) and dying INS-1 cells (PI-AnnexinV+) from live INS-1 cells (PI-AnnexinV).

## RNA Seq

Total RNA from ~1 × 10^6 INS-1 cells was isolated using the RNAeasy mini kit (Qiagen) according to the manufacturer's protocol. RNA integrity was assessed using Ribogreen to determine total mass and Fragment Analyzer was used to determine RNA integrity. All samples passed quality control, with RNA integrity scores are ranging from 7.7 to 10. Samples were submitted to Genewiz and RNA was amplified via the NuGEN Ovation RNA-Seq System V2 prior to RNA sequencing. Each sample was sequenced on a HiSeq2500 instrument (Illumina) at a depth of 35–40 million, 150 bp paired-end reads, that were single-indexed per lane. Raw counts were processed for DGE analysis using the edgeR R package (version 4.2), using a minimum fold change of 1.2. Six DGEs were conducted to reflect each possible comparison of the 4 groups. The resulting lists of differentially expressed genes were fed into the ViSEAGO R package (version 1.8.0). ViSEAGO carries out a data mining of biological functions and establishes links between genes involved in the study facilitating functional GO analysis of complex experimental design with multiple comparisons[29]. For human studies, β-cells from dispersed human islets were transduced with an adenovirus expressing ZsGreen driven by a MIP-miniCMV promoter and harvested by fluorescence-activated cytometric sorting (FACS Aria II) as described previously[22,66]. The β-cell fraction was confirmed to be >92% pure by immunolabeling of sorted cells with insulin, by qRT-PCR and by RNAseq[66]. RNA sequencing, alignment and feature quantitation was as previously described[22]. The datasets generated during and/or analyzed during the current study are available in the GEO repository as accession number GSE197864.

## qPCR and PCR

mRNA was isolated using the Qiagen RNAeasy mini kit for INS-1 cells, or for islets using the Qiagen RNAeasy mini kit. cDNA was produced using the Promega m-MLV reverse transcriptase. qPCR was performed on the QuantStudio5 using Syber-Green (BioRad) and analysis was performed using the ΔΔCt method. PCR for genotyping was performed using standard methods. Primer sequences are shown in Supplementary Table 1.

## Western blotting and chromatin immunoprecipitation

Cells were lysed in RIPA buffer (Thermo-Fisher) with protease inhibitors (Roche). Lysates were sonicated for 20 s on ice and centrifuged at $10,000 \times g$ for 5 min. Lysates were then placed in a 95 °C for 5 min with Laemmli loading buffer. A total of 40 µg protein extract/well was loaded on 7.5% SDS polyacrylamide gels. Western blots were carried out using antibodies described in Supplementary Table 2, and the blots were scanned with the LI-COR laser-based image detection method as previously described[73]. Chromatin immunoprecipitation was performed as previously described[21].

## Statistics

All studies were performed with a minimum of three independent repletion. Data presented in this study as means ± standard error of the mean (SEM). Statistical analysis was performed using Two-way ANOVA on GraphPad (Prism) V9.2.

## Reporting summary

Further information on research design is available in the Nature Research Reporting Summary linked to this article.

## Data availability

The datasets generated during and/or analyzed during the current study are available in the GEO repository as accession number GSE197864. Source data are provided with this paper.

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

## Acknowledgements

D.K.S., R01DK108905, R01DK114338, R01DK130300; J.C.L., R01DK12140; A.F.S., R01DK116873, R01DK125285, R01DK129196; A.G.-O., R01DK125285, R01DK105015, R01DK126450; A.A., Charles H. Revson Foundation (grant no. 18-25), Sweden-America Foundation (Ernst O Eks fond), Swedish Society for Medical Research (SSMF); S.S., American Diabetes Association Pathway to Stop Diabetes Grant ADA #1-17-ACE-31, R01NS097184, OT2OD024912, and R01DK124461, Department of

Defense (W81XWH-20-1-0345, W81XWH-20-1-0156); D.H., R01AG026518 and R01AI093637, Juvenile Diabetes Research Foundation Career Development Award 2-2007-240; (B.D.) T32 DK007792; MAH, R01DK100425. We thank the Boston Nutrition Obesity Research Center for generation of mice. We thank the Flow Cytometery, Microscopy, Mouse Genetics and Gene Targeting, the Biorepository and Pathology Cores of Icahn School of Medicine at Mount Sinai. We also thank the Human Islet and Adenovirus Core of the Einstein-Mount Sinai Diabetes Research Center (DK-020541) for generation of adenoviruses and islet transplantation services. We thank Pedro Herrera and Daniel Oropeza for useful discussions.

## Author contributions

Conceptualization, L.S.K., and D.K.S.; Methodology, L.S.K., A.A., S.A.S., B.D., D.H., P.W., L.D., and M.A.H.; Investigation, L.S.K., G.B., P.Z., A.K., S.B-A., L.B.H., N.G-B., E.K., and M.T.; Formal analysis, L.L.; Resources, N.G-B., J.C.L., L.D., and M.A.H., Writing—original draft, L.S.K., and D.K.S; Writing-review and editing, L.S.K., D.S.K., A.F.S., and A.G-O., Supervision and funding acquisition, S.A.S., A.F.S., D.H., J.C.L., M.A.H., A.G.-O., and D.K.S.

## Competing interests

The authors declare no competing interests.
