## [Peer Review File · Nature Communications]

Reviewers' Comments:

Reviewer #1:

Remarks to the Author:

Montreal, 23th of January 2022

The manuscript "Maladaptive Positive Feedback Production of ChREBP β Underlies Glucotoxic β -Cell Failure" by Liora S. Katz et al provides a comprehensive overview of the roles and biology of the transcription factor carbohydrate responsive element binding protein (ChREBP) in the pancreatic beta cells.

Since the discovery of the truncated constitutively nuclear ChREBPbeta isoform in the adipose tissue (Herman et al, Nature 2012), it was becoming important to ascertain the respective roles of ChREBPalpha and ChREBPbeta in the pancreatic beta cells, since ChREBP had long been postulated to play a role in glucotoxicity in this tissue.

In the pancreatic beta cells, ChREBP inhibits transcription of hypoxia-inducible factor-1B (HIF-1B/ARNT) (Noordeen et al, Diabetes 2009), a regulator of the response to hypoxic stress, loss of which has been shown to cause pancreatic islet dysfunction in type 2 diabetes mellitus (Gunton et al, Cell 2005). ChREBP also induces expression of fatty acid synthase, and thioredoxin-interacting protein (TxNIP), which lead to lipid accumulation, reduced insulin secretion, oxidative stress and beta-cell apoptosis (daSilvaXavier et al, JLR 2006; Noordeen et al, Diabetes 2012; Pongvarin et al, Diabetologia 2012).

In this manuscript, using a combination of isoform specific antibodies, CRISPR/Cas9 engineered pancreatic beta cell lines to allow distinction between ChREBPalpha and beta, two different models of ChREBPbeta knock out mice, two models of ChREBPbeta knock in mice, and a rescue by ChREBPalpha, the authors have convincingly provided evidence that ChREBPbeta is one of the main integrator and effector of beta cell loss during the progression of type 2 diabetes (T2D). The authors have also validated their data in human pancreatic islets, firstly ex vivo in islets isolated from non-diabetic donors and from T2D donors, FACS sorted to enrich in beta cells after transduction with RIP-ZsGreen expressing adenoviruses, but also in vivo by transplanting human islets, overexpressing either Cre or ChREBPalpha, under the renal capsule of immunocompromised mice rendered diabetic by streptozotocin injections prior to transplantation.

It is a technical tour de force and the data are very convincing. Interestingly, ChREBPalpha has a protective effect and increase insulin secretion, hence the apparent conflicting results reported in the literature by various groups. ChREBPbeta on the other hand becomes active under sustained hyperglycemia and high fat feeding and this results in beta cell death and diabetes.

Hopefully, the data presented in this paper should fast-track efforts by the pharmacological industry to develop drugs that would specifically target this pathway to protect the pancreatic beta cells. This is now urgently needed given the current epidemics of obesity and T2D and the devastating consequences of diabetes complications in an increasingly younger population.

General comments: Although the paper is very well written, the methods section and the figure legends would benefit from careful revision by a more experienced author. Quite a lot of details are missing. This is particularly true for the supplemental figure legends, for example, in SF2: A and B not described; in SF3: there is mention of A and B, but only A is shown, etc.

One major point that has not been addressed properly is the nature of the glucose signal that stimulates the translocation of ChREBPalpha in the pancreatic beta cell. In figure legend 1, it is mentioned that a metabolite of glucose, most likely G6P, causes a conformational change, exposing the N-terminal activation domain. No reference were cited but the evidence for this in the pancreatic beta cells are flawed (i.e. paper by Li et al, BBRC 2010). In the pancreatic beta cell, contrary to the situation in hepatocytes, glucose phosphorylation into G6P is accompanied by an increase in intracellular calcium, blockage of which by diazoxide totally inhibits the response of the L-PK promoter to glucose (da Silva Xavier et al, JBC, 2000).

In the paper by Li et al, no attempt were made to measure intracellular calcium in INS832/13 cells, but it is well known that inhibiting the formation of G6P, be it by inhibiting hexokinase or glucokinase, will also prevent glucose-induced rise in intracellular calcium (Idevall-Hagren & Tengholm Seminars in Cell and Developmental Biology, 2020). The other argument used in this paper to sell the G6P metabolite hypothesis is the use of the glucose analogue, 2-deoxyglucose (2-DG), which can be phosphorylated by hexokinase, but not further metabolized. It is well known that 2-DG will cause a drop in intracellular ATP (Foufelle F, Journal of Biological Chemistry 1992) which in turn will increase intracellular calcium (Haworth et al. Circ Res 1987, Spivey et al. JCI 1993, Bright et al. 1995). In the paper by Noordeen et al, Diabetes 2012, it is well demonstrated that increasing intracellular calcium alone was sufficient to provoke a nuclear translocation of ChREBP, making an increase in Ca²⁺, not a metabolite of glucose, the nature of the glucose signal in the excitable pancreatic beta cell (review in Leclerc et al, J of Endocr, 2012).

Dr Isabelle Leclerc, MD, PhD, FRCP (Canada), MRCP (London)
Professor of Clinic, Faculty of Medicine, University of Montreal
Visiting Professor, Department of Medicine, Imperial College London

Department of Endocrinology
Centre Hospitalier de l'Université de Montréal (CHUM)
850, rue Saint-Denis, Pavillon B, 10-8074
Montreal (Quebec)
H2X 0A9, Canada

isabelle.leclerc.chum@ssss.gouv.qc.ca

Reviewer #2:

Remarks to the Author:

This manuscript provides definitive genetic tests for the unique roles of ChREBP beta in pancreatic beta cells. The authors have made and characterized a series of highly informative tools and reagents to distinguish ChREBP alpha and beta isoforms, which they have used to rigorously corroborate their conclusions. The data are of high quality and the manuscript is significant for its in vivo studies using multiple genetic models to address the role of these isoforms in beta cell development as well as beta cell mass adaptation to HFD and decompensation in response to chronic hyperglycemia. The data show that ChREBP β is necessary for β -cell mass expansion in response to HFD, yet, prolonged expression of ChREBP β leads to β -cell death, which is rescued by ChREBP α or Nrf-2. The in vivo findings of a positive feedback loop are original and important to advance our understanding of the compensatory responses of these transcriptional regulators. Overall, this manuscript is an important contribution to the field.

Comments:

1. The authors should consider expanding the discussion on how the ratio of beta and alpha may be influenced by mechanisms beyond transcription, including differences in nuclear retention of the two isoforms. Is the nuclear export of the alpha isoform dynamically regulated? Are there differences in mRNA stability and translation rates of the two isoforms in response to diabetes related stress signals?
2. The discussion can also benefit from comments on the extent of differences in beta cell transcriptional programs downstream of ChREBP alpha vs beta.
3. The islet transplantation results are very convincing, however could the images in Figure 3L be used to quantify nuclear localization of ChREBP to help draw correlations with the degree of changes seen in blood glucose levels.
4. Does the same nuclear localization pattern occur at physiologic glucose concentrations? This could be tested in any cell system but might be more feasible in primary islets, which have normal glucose sensing/responses and could be assessed at 11mM glucose. This could be further informative as it's the same glucose concentrations used in the RNA expression studies.
5. In Fig. 7D-G, please provide the scale bar heatmap values, only "min" and "max" is indicated but there should be numerical values.

6. Are the images in Figure 2B adjustable to lower the background signal? This could make it easier to see the true ChREBP alpha signal.

We thank the reviewers for their enthusiasm and positive comments such as: “It is a technical tour de force and the data are very convincing”; “Hopefully, the data presented in this paper should fast-track efforts by the pharmacological industry to develop drugs that would specifically target this pathway to protect the pancreatic beta cells”; “Overall, this manuscript is an important contribution to the field.” We also thank the reviewers for their many helpful suggestions. We address each of their comments below in blue. We believe that now, with the help of the reviewers’ insight, the paper is even stronger.

REVIEWER COMMENTS

Reviewer #1 (Remarks to the Author):

Montreal, 23th of January 2022

The manuscript “Maladaptive Positive Feedback Production of ChREBP β Underlies Glucotoxic β -Cell Failure” by Liora S. Katz et al provides a comprehensive overview of the roles and biology of the transcription factor carbohydrate responsive element binding protein (ChREBP) in the pancreatic beta cells.

Since the discovery of the truncated constitutively nuclear ChREBP β isoform in the adipose tissue (Herman et al, Nature 2012), it was becoming important to ascertain the respective roles of ChREBP α and ChREBP β in the pancreatic beta cells, since ChREBP had long been postulated to play a role in glucotoxicity in this tissue.

In the pancreatic beta cells, ChREBP inhibits transcription of hypoxia-inducible factor-1B (HIF-1B/ARNT) (Noordeen et al, Diabetes 2009), a regulator of the response to hypoxic stress, loss of which has been shown to cause pancreatic islet dysfunction in type 2 diabetes mellitus (Gunton et al, Cell 2005). ChREBP also induces expression of fatty acid synthase, and thioredoxin-interacting protein (TxNIP), which lead to lipid accumulation, reduced insulin secretion, oxidative stress and beta-cell apoptosis (daSilvaXavier et al, JLR 2006; Noordeen et al, Diabetes 2012; Pongvarin et al, Diabetologia 2012).

In this manuscript, using a combination of isoform specific antibodies, CRISPR/Cas9 engineered pancreatic beta cell lines to allow distinction between ChREBP α and β , two different models of ChREBP β knock out mice, two models of ChREBP β knock in mice, and a rescue by ChREBP α , the authors have convincingly provided evidence that ChREBP β is one of the main integrator and effector of beta cell loss during the progression of type 2 diabetes (T2D). The authors have also validated their data in human pancreatic islets, firstly ex vivo in islets isolated from non-diabetic donors and from T2D donors, FACS sorted to enrich in beta cells after transduction with RIP-ZsGreen

expressing adenoviruses, but also in vivo by transplanting human islets, overexpressing either Cre or ChREBPalpha, under the renal capsule of immunocompromised mice rendered diabetic by streptozotocin injections prior to transplantation.

It is a technical tour de force and the data are very convincing. Interestingly, ChREBPalpha has a protective effect and increase insulin secretion, hence the apparent conflicting results reported in the literature by various groups. ChREBPbeta on the other hand becomes active under sustained hyperglycemia and high fat feeding and this results in beta cell death and diabetes.

Hopefully, the data presented in this paper should fast-track efforts by the pharmacological industry to develop drugs that would specifically target this pathway to protect the pancreatic beta cells. This is now urgently needed given the current epidemics of obesity and T2D and the devastating consequences of diabetes complications in an increasingly younger population.

Thank you for your kind words and valuable input. We agree! Prevention of β -cell loss is an important unmet need in diabetes management. Our data, built on work from many other labs, supports ChREBP β as a therapeutic target for prevention of gluco(lipo)toxicity.

General comments: Although the paper is very well written, the methods section and the figure legends would benefit from careful revision by a more experienced author. Quite a lot of details are missing. This is particularly true for the supplemental figure legends, for example, in SF2: A and B not described; in SF3: there is mention of A and B, but only A is shown, etc.

Thank you. We have carefully revised the Method and Legend sections, and have largely re-written the supplementary figure legends.

One major point that has not been addressed properly is the nature of the glucose signal that stimulates the translocation of ChREBPalpha in the pancreatic beta cell. In figure legend 1, it is mentioned that a metabolite of glucose, most likely G6P, causes a conformational change, exposing the N-terminal activation domain. No reference were cited but the evidence for this in the pancreatic beta cells are flawed (i.e. paper by Li et al, BBRC 2010). In the pancreatic beta cell, contrary to the situation in hepatocytes, glucose phosphorylation into G6P is accompanied by an increase in intracellular calcium, blockage of which by diazoxide totally inhibits the response of the L-PK promoter to glucose (da Silva Xavier et al, JBC, 2000).

In the paper by Li et al, no attempt were made to measure intracellular calcium in INS832/13 cells, but it is well known that inhibiting the formation of G6P, be it by inhibiting hexokinase or glucokinase, will also prevent glucose-induced rise in intracellular calcium (Idevall-Hagren & Tengholm Seminars in Cell and

Developmental Biology, 2020). The other argument used in this paper to sell the G6P metabolite hypothesis is the use of the glucose analogue, 2-deoxyglucose (2-DG), which can be phosphorylated by hexokinase, but not further metabolized. It is well known that 2-DG will cause a drop in intracellular ATP (Foufelle F, Journal of Biological Chemistry 1992) which in turn will increase intracellular calcium (Haworth et al. Circ Res 1987, Spivey et al. JCI 1993, Bright et al. 1995). In the paper by Noordeen et al, Diabetes 2012, it is well demonstrated that increasing intracellular calcium alone was sufficient to provoke a nuclear translocation of ChREBP, making an increase in Ca^{2+} , not a metabolite of glucose, the nature of the glucose signal in the excitable pancreatic beta cell (review in Leclerc et al, J of Endocr, 2012).

Many thanks. We agree with your assessment and agree that “most likely G6P” is an oversimplification and not completely supported by the published data in beta cells. To address this, we have removed “a metabolite of glucose, most likely glucose 6 phosphate (G6P)” from legend of figure 1 and the Discussion, and have modified the figure and legend accordingly. We also added calcium in the figure and legend. In the Discussion, we have modified the narrative to address the lack of consensus for exactly how ChREBP α is activated, and have included Ca^{++} -regulated translocation, as follows: “While transcriptional regulation and localization of the two splice isoforms, as demonstrated in this study seem to be of high importance for the regulation of ChREBP and activity, there are additional mechanisms that have been previously shown to tightly regulate ChREBP location and activity. ChREBP is regulated by carbohydrate metabolites and other metabolic signals (Abdul-Wahed et al., 2017; Agius et al., 2020), including Ca^{++} flux, which dissociates its binding to sorcin allowing nuclear translocation in beta cells (Noordeen et al., 2012), and several posttranslational modifications [reviewed in (Katz et al., 2021)], which may affect its stability and the binding of co-factors and co-activators (Bricambert et al., 2018; Lane et al., 2019).”

Dr Isabelle Leclerc, MD, PhD, FRCP (Canada), MRCP (London)
Professor of Clinic, Faculty of Medicine, University of Montreal
Visiting Professor, Department of Medicine, Imperial College London

Department of Endocrinology
Centre Hospitalier de l'Université de Montréal (CHUM)
850, rue Saint-Denis, Pavillon B, 10-8074
Montreal (Quebec)
H2X 0A9, Canada

isabelle.leclerc.chum@sss.gouv.qc.ca

Reviewer #2 (Remarks to the Author):

This manuscript provides definitive genetic tests for the unique roles of ChREBP beta in pancreatic beta cells. The authors have made and characterized a series of highly informative tools and reagents to distinguish ChREBP alpha and beta isoforms, which they have used to rigorously corroborate their conclusions. The data are of high quality and the manuscript is significant for its in vivo studies using multiple genetic models to address the role of these isoforms in beta cell development as well as beta cell mass adaptation to HFD and decompensation in response to chronic hyperglycemia. The data show that ChREBP β is necessary for β -cell mass expansion in response to HFD, yet, prolonged expression of ChREBP β leads to β -cell death, which is rescued by ChREBP α or Nrf-2. The in vivo findings of a positive feedback loop are original and important to advance our understanding of the compensatory responses of these transcriptional regulators. Overall, this manuscript is an important contribution to the field.

Many thanks for your kind words.

Comments:

1. The authors should consider expanding the discussion on how the ratio of beta and alpha may be influenced by mechanisms beyond transcription, including differences in nuclear retention of the two isoforms. Is the nuclear export of the alpha isoform dynamically regulated? Are there differences in mRNA stability and translation rates of the two isoforms in response to diabetes related stress signals?

Thank you for this suggestion. We have added a paragraph to the discussion: "While transcriptional regulation and localization of the two splice isoforms, as demonstrated in this study seem to be of high importance for the regulation of ChREBP and activity, there are additional mechanisms that have been previously shown to tightly regulate ChREBP location and activity. ChREBP is regulated by carbohydrate metabolites and other metabolic signals (Abdul-Wahed et al., 2017; Agius et al., 2020), including Ca⁺⁺ flux, which dissociates its binding to sorcin allowing nuclear translocation in beta cells (Noordeen et al., 2012), and several posttranslational modifications [reviewed in (Katz et al., 2021)], which may affect its stability and the binding of co-factors and co-activators (Bricambert et al., 2018; Lane et al., 2019). Additionally, nuclear retention of ChREBP's mRNA (Bahar Halpern et al., 2015), and sequestration of ChREBP's heterodimer partner, Mix, in lipid droplets play important roles in regulating ChREBP activity (Mejhert et al., 2020). Furthermore, there are other mechanisms that have not been fully explored that may regulate the ratio of ChREBP isoform activity during increasing metabolic stress including differential mRNA and protein stability and differential protein translation rates. Clearly, a transcription factor that plays such an important role in beta cell function, proliferation, and apoptosis should be tightly controlled."

2. The discussion can also benefit from comments on the extent of differences in beta cell transcriptional programs downstream of ChREBP alpha vs beta.

Thank you for this suggestion. We have added to the Discussion the following: “Collectively, these results suggest that ChREBP β and ChREBP α drive different transcriptional programs. On the one hand, overexpression of both ChREBP α and ChREBP β result in decreased expression of β -cell markers [(da Silva Xavier et al., 2010) and this study]. On the other hand, overexpression of ChREBP β leads to hyper-expression of Txnip and cell death, whereas overexpression of ChREBP α activates the Nrf2 antioxidant pathway via unknown mechanism, resulting in enhanced glucose-stimulated proliferation and protection from ChREBP β -mediated cell death [(Kumar et al., 2018) and this study]. The identity of the different gene targets of the two ChREBP isoforms and the mechanisms by which they are regulated are areas of active investigation.”

3. The islet transplantation results are very convincing, however could the images in Figure 3L be used to quantify nuclear localization of ChREBP to help draw correlations with the degree of changes seen in blood glucose levels.

Thank you for this suggestion. We added a panel in Figure 3M. We found a strong correlation between nuclear localization of ChREBP β and blood glucose levels.

4. Does the same nuclear localization pattern occur at physiologic glucose concentrations? This could be tested in any cell system but might be more feasible in primary islets, which have normal glucose sensing/responses and could be assessed at 11mM glucose. This could be further informative as it's the same glucose concentrations used in the RNA expression studies.

Thank you for this suggestion. We added Suppl. Fig 5C, D to show ChREBP β localization in isolated mouse islets at 6, 11 and 20 mM glucose. It is interesting to note that in all our model systems, more physiological concentrations (11 mM) lead to modest, but significant increases in the nuclear localization of ChREBP β , but pathophysiological concentrations (20 mM) lead to robust nuclear staining of ChREBP β .

5. In Fig. 7D-G, please provide the scale bar heatmap values, only “min” and “max” is indicated but there should be numerical values.

We now provide a scale. Thank you.

6. Are the images in Figure 2B adjustable to lower the background signal? This could make it easier to see the true ChREBP alpha signal.

Thank you. We replaced the images in Fig. 2B with better quality images.

Reviewers' Comments:

Reviewer #1:

Remarks to the Author:

The authors have responded satisfactorily to my comments. As previously stated in my initial review, this is an important paper that will give tools to allow translational research to focus on preserving beta cell mass and function during the progression of insulin resistance and hyperglycaemia in Type 2 diabetes and obesity.

Reviewer #2:

Remarks to the Author:

The authors have fully addressed my previous comments.

We thank the reviewers for their supportive and constructive comments. Our response is below in blue.

REVIEWERS' COMMENTS

Reviewer #1 (Remarks to the Author):

The authors have responded satisfactorily to my comments. As previously stated in my initial review, this is an important paper that will give tools to allow translational research to focus on preserving beta cell mass and function during the progression of insulin resistance and hyperglycaemia in Type 2 diabetes and obesity.

Thank you!

Reviewer #2 (Remarks to the Author):

The authors have fully addressed my previous comments.

Thank you!